# Biophilic classroom environments on stress and cognitive performance: A randomized crossover study in virtual reality (VR)

Jicheng You[1,2], Xinyi Wen[1,3,4], Linxin Liu[5,6], Jie Yin[7], John S. Ji[1,3,6,8,9] *

**1** Center for the Study of Contemporary China, Duke Kunshan University, Jiangsu, China, **2** Department of Biomedical Engineering, Columbia University, New York, NY, United States of America, **3** Nicholas School of the Environment, Duke University, Durham, NC, United States of America, **4** Pratt School of Engineering, Duke University, Durham, NC, United States of America, **5** School of Medicine, Tsinghua University, Beijing, China, **6** Vanke School of Public Health, Tsinghua University, Beijing, China, **7** Key Laboratory of Ecology and Energy-Saving Study of Dense Habitat, College of Architecture and Urban Planning, Tongji University, Shanghai, China, **8** Global Health Research Center, Duke Kunshan University, Jiangsu, China, **9** Environmental Research Center, Duke Kunshan University, Jiangsu, China

Ↄ These authors contributed equally to this work.

* johnji@tsinghua.edu.cn

**Data Availability Statement:** The code for data cleaning and statistical analysis that support the findings of the study are available in GitHub (https://github.com/Carl-J/Data-processing_VR)

## Abstract

The emerging Metaverse will likely increase time expenditure in indoor virtual environments, which could impact human health and well-being. The biophilia hypothesis suggests that humans have an innate tendency to seek connections with the natural world and there is increasing evidence that biophilic design such as the incorporation of green plants can yield health benefits. Recently, virtual reality (VR) has been used to regulate stress and improve overall wellness, particularly by incorporating natural settings. In this randomized crossover study, we designed five virtual classroom scenes with different biophilic elements and turbidity in VR and investigated whether the visual stimulations can affect the stress levels and cognitive functions of 30 young adults from a university in China. We measured their physiological indicators of stress reaction by wearable biomonitoring sensors (blood pressure (BP), heart rate (HR), heart rate variability (HRV), and skin conductance level (SCL)), conducted verbal cognitive tests on attention and creativity, and evaluated subjective/perceived (self-reported) stress levels and connection with nature. Albeit our results suggested no significant change in physiological stress reactions or cognitive functions induced by the biophilic and turbid interventions in VR, the addition of biophilic elements in the Metaverse could benefit students' health due to significantly decreased perceived stress levels and increased connections with nature.

## Introduction

Globally, there's a noticeable decline in access to natural settings, driven in part by the shift towards sedentary learning and work habits. This is profoundly evident among students,

with the identifier https://doi.org/10.5281/zenodo.8303723. The source data for each participant is provided in Supporting Information S1 Data.

**Funding:** J.S.J. received funding from the Center for the Study of Contemporary China (CSCC) at Duke Kunshan University for an Undergraduate Research Grant for the academic year 2020-2021 (https://www.dukekunshan.edu.cn/cscc/) and funding from the National Natural Sciences Foundation of China (Grant number: 82250610230). The funders had no role in study design, data collection, and analysis, decision to publish, or preparation of the manuscript.

**Competing interests:** The authors have declared that no competing interests exist.

many of whom have and will continue to allocate substantial time to remote learning. Additionally, with the advent of the Metaverse, particularly accentuated by Facebook's rebranding [1] the trend of spending more time in both physical indoor settings and virtual indoor environments is poised to grow. Regrettably, a number of studies, predominantly reliant on questionnaires, have revealed that students from a vast range of grade levels across numerous countries have reported elevated levels of stress, anxiety, and depression during periods of online learning [2–6]. Heightened use of smart devices and increased screen time have been linked to an array of stress-induced symptoms [3, 7] and poorer academic performance [6].

The *biophilia hypothesis*, popularized by biologist E.O. Wilson in 1984, suggests that humans have an innate tendency to seek connections with the natural world [8–10]. Two complementary theories, *Stress Recovery Theory* (SRT) and *Attention Restoration Theory* (ART), support the hypothesis. SRT indicates that the activated parasympathetic nervous system by natural elements leads to stress reduction [8], while ART suggests that natural environments are restorative by capturing involuntary attention [11]. Population epidemiology studies and experiments documented exposure to *outdoor* nature (e.g., greenspace) can positively affect human health and well-being in multiple ways: reduced stress level, improved mental health and cognition, lower mortality rates [12–16], and enhanced immune functions [17–20]. In contrast, the health impact of *indoor* environmental quality has been examined only recently. Most research focused on *negative* factors impacting human health such as poor indoor air quality and materials with toxic chemicals. Currently, only a few studies investigated positive attributes like biophilic design which incorporates natural elements in indoor spaces [21].

There is emerging evidence that biophilic design in simulated environment can yield health benefits. For example, indoor plants could be conducive to stress-reduction and attention restoration [22, 23], viewing nature through a window helped patients reduce recovery time [24–26], and natural light could raise feelings of vitality and quality of life [27, 28]. In recent years, the use of virtual reality (VR) systems is more commonly implemented to improve human health. Virtual environments that aim to reduce stress often incorporate natural settings because of their well-researched and proven ability to regulate stress and improve overall wellness [29]. Combined biophilic design elements using virtual reality (VR) by Yin et al. [30] demonstrated that bringing nature into virtual indoor workspace has clear benefits to the health outcomes, including physiological stress reductions and cognitive function (attention and creativity) improvements. Combining VR, eye-tracking, and wearable biomonitoring sensors, their studies supported the dominance of visual senses in creating perceptions [31] and provided a potential tool for objective virtual exposure assessment (e.g., measuring physiological stress reactions such as blood pressure and heart rate).

To explore the design of study environment in the Metaverse for students and examine their health responses to the biophilic elements, we used VR and wearable biomonitoring sensors to quantify the impacts of both positive and negative factors in built virtual environments on short-term health (i.e., stress reduction and cognitive function improvement within minutes or hours after exposure) of university students. Specifically, the study investigated the immediate physiological and cognitive responses (i.e., during the experiment session) to five virtual scenes of university classroom (same classroom) with different biophilic elements and turbidity (i.e., hazy outdoor view) which mimics air pollution visually. Although hazy windows are not the opposite to biophilic design, a study found that photos of gray cityscape caused by particle pollution could impede human stress recovery [32]. We used a randomized crossover design to achieve the same statistical power with fewer participants [33, 34], i.e., participants served as their own control group with physiological and cognitive responses being repeatedly measured. All participants experienced five different virtual scenes randomly during a single experiment session. We hypothesized that participants would experience physiological stress

reduction and cognitive function improvement after exposures to various biophilic classroom environments in VR, while the turbid environment would negatively influence participants' short-term health, or immediate impact after exposure. Specifically, we also wanted to explore the influence from various biophilic designs depending on the content of indoor and outdoor elements.

## Materials and methods

### Study population

The study recruited 30 participants via advertisement and social networks platforms. Two rounds of recruitment took place in July and August 2021. Inclusion criteria were 18–30 years old, free from hypertension and heart diseases, and not taking medicine or therapy for stress recovery or relief (S1A File). Eligible participants were invited to a classroom in the Innovation Building at Duke Kunshan University (DKU) to participate in the study and were compensated with a gift valued at RMB 50 (USD $7.28).

### Ethics statement

All participants provided informed written consent. The study protocol was reviewed and approved by the DKU Institutional Review Board (Approval number: FWA00021580).

### Environment simulation

In this study, we chose three biophilic patterns from the conceptual framework for biophilic design [35] because (1) they relate to indoor classroom design, (2) they can be vividly simulated in VR, and (3) they are suitable for short exposure time (e.g., 5 minutes). It has been proved that these patterns can induce significant changes in people's physiological responses and cognitive functions [30, 36]. Specifically, the patterns of "Visual connection to nature" and "Dynamic and diffuse light" were combined to represent *Nature in the Space*, which included potted plants, trees, sky, clouds, and access to natural light and shadow. We used the pattern of "Material connection with nature" to represent *Natural Analogues*, including wooden floors and ceilings.

With these patterns deposited, we simulated five virtual indoor classroom environments based on a real classroom at DKU (Fig 1A), including one non-biophilic environment as control (Fig 1B) and four biophilic interventions. The biophilic intervention refers to the virtual scene with biophilic design elements. Intervention 1 is Indoor Green (Fig 1C) where the classroom is decorated with green plants and natural materials. Intervention 2, named Outdoor Green (Fig 1D), incorporates outdoor natural view and daylight into indoor space through windows. In Intervention 3, Turbid Outdoor Green (Fig 1E), the outdoor natural view in Intervention 2: Outdoor Green is blocked with visual turbidity, which is intended to simulate air pollution. The biophilic elements in Indoor Green and Outdoor Green are combined in Intervention 4 –Combination (Fig 1F).

To accommodate the quality-consuming task for VR, we purchased and assembled CPU AMD R7 5800X, mainboard ASUS TUF GAMING B550M-plus, and GPU NVIDIA GeForce GTX2070. The initial 3D virtual classroom model building was done in Rhino 7 software by a professional vendor. 3D models were then modified and rendered by Simlab Composer 10 and stored locally. We chose HTC VIVE CE (Fig 2) as the ideal headset because of its high resolution, 90Hz refresh rate, and 110˚ field of view. The interface platform was based on SteamVR since the headset and controllers used SteamVR tracking system. The two standing base stations can accurately define any area within 11'5"x11'5" room-scale and track the

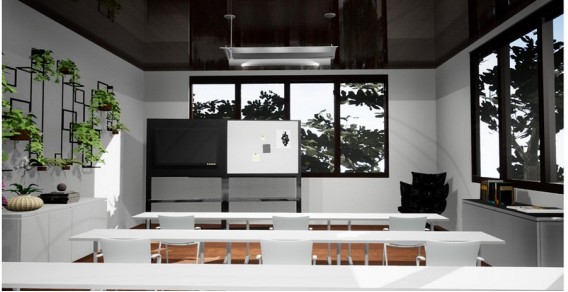

**Fig 1. 3D models of the five virtual classroom environments and the laboratory space.** (A) The classroom for conducting the experiment at Duke Kunshan University. (B) Control: Non-biophilic classroom. (C) Intervention 1: Indoor green–green plants and natural materials. (D) Intervention 2: Outdoor green–natural view and daylight into indoor space through windows. (E) Intervention 3: Turbid outdoor green–blocked outdoor view with visual turbidity (same classroom view as intervention 2). (F) Intervention 4: Combination–biophilic elements from indoor green and outdoor green.

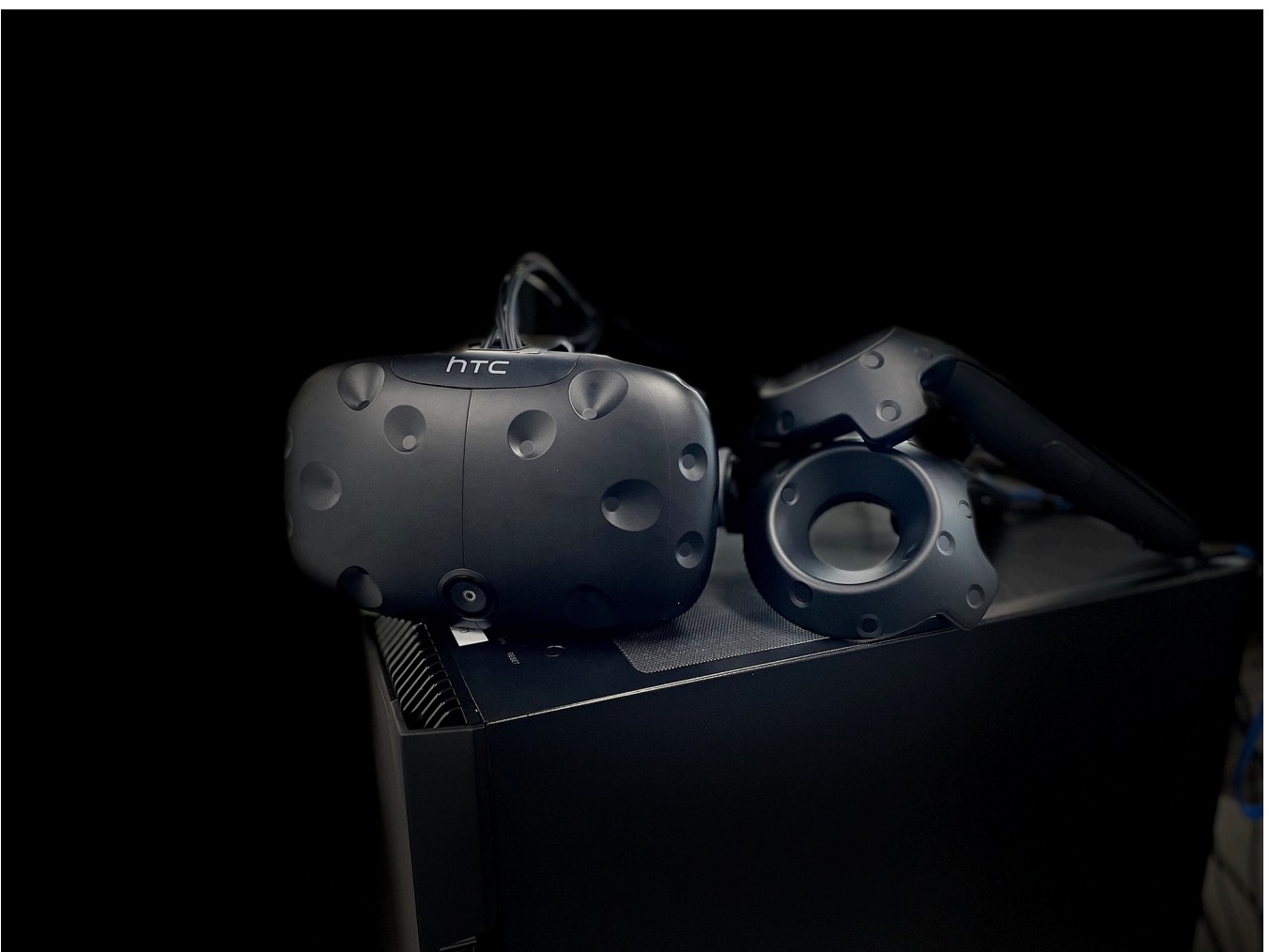

**Fig 2. HTC VIVE CE headset and controllers.** Photo credit: Xinyi Wen.

position (including altitudes) of players synchronously. Additionally, the multifunction track-pad on the wireless controller enabled the players to teleport freely within the virtually defined area. Since the modeling was rendered by Simlab Composer, we used compatible Simlab VR viewer during experiment to display the stored scenes. Most simulated environments enjoyed a fluent refresh rate higher than approximately 60 Hz. Indoor Green, however, presented a lower fresh rate than other scenes when projected on the headset probably due to the higher quantity of plants implanted within it.

## Physiological indicators of stress reaction

We used four physiological indicators to assess participants' acute stress reaction (blood pressure (BP), heart rate (HR), heart rate variability (HRV), and skin conductance level (SCL), which were obtained either by wearable biomonitoring sensors or calculation.

Blood pressure is correlated with the force the blood exerts on the vascular walls [37] at stress reactions [38]. We attached the Omron J760 blood pressure monitor to the upper right arm of participants to measure their systolic blood pressure (SBP) and diastolic blood pressure (DBP) (Omron Healthcare Inc.). The SBP refers to the force exerted on arteries during a

heartbeat while the DBP signifies the pressure in arteries while heart is in a resting state between beats. Same as previous studies, we analyzed both numbers for the stress reaction since they constitute a complete measurement of BP [39].

Closely correlates with BP, HR is a measurement for heartbeat frequency (bpm) [40]. The electrodermal activity sensor, Shimmer3 GSR+Unit was attached to participants' left earlobe to collect the Photoplethysmography (PPG) data. The ConsensysPro software automatically converted the PPG signals into time series HR (beats per minute). HRV describes the dynamic interaction between parasympathetic and sympathetic branches of the autonomic nervous system (ANS) [41], which responded to stress induced by various methods in most studies [42] and is related to physiological stress responses [41]. We chose root mean square of successive differences between normal heartbeats (RMSSD) (milliseconds [ms]) for short-term HRV measurement [43]. It was later calculated using the formula

$$RMSSD = \sqrt{\frac{1}{N-1}\sum_{i=1}^{N}(RR_{i+1} - RR_i)^2}$$

for each environment, and higher values indicate lower stress level [44]. N represents the number of data points collected in the measuring period, while $i$ means the data point examined. RR represents inter-beat interval (IBI) which is automatically calculated and exported by ConsensysPro Software. SCL is widely used as an indicator for physiological stress associated with exposure to natural environments [33, 45–47]. It measures the electrodermal activity in the sweat glands that is controlled by the autonomic nervous system [8]. Shimmer3 GSR+Unit was worn on two fingers (middle and ring finger) on participants' left hands to collect the SCL (μS). The PPG and SCL data collections were all conducted at 128 Hz.

Momentary measures of BP were conducted immediately before and after the exposure to each virtual scene, while HR and SCL were measured continuously throughout the experiment. HRV was manually calculated based on the measured HR by the formula above. To supplement those physiological stress measures, we also collected self-reported stress levels during the experiment. Participants were asked to rate their current stress levels immediately after the exposure to each scene with a range from 1 to 5 (no decimal) where 1 refers to "not stressful" and 5 denotes "extremely stressful." We calculated the difference in the stress levels between the biophilic interventions and the control non-biophilic scene.

### Cognitive function assessment

Empirical research suggested that VR can potentially facilitate better cognitive function assessments than traditional methods [48–50]. In this study, we conducted two cognitive tests upon intervention exposures to investigate VR simulation's effect on convergent (attention) and divergent cognitive functions (creativity). We chose attention and creativity because they can be assessed through validated cognitive tests and previous studies showed that they could potentially be influenced by short-term environmental exposures in VR [30, 33]. To minimize the visual interferences in VR, we delivered the two tests verbally. Specifically, the participants remained in the biophilic intervention scene and sat still while one investigator (the same person for all the participants) verbally asked the questions and the participants responded verbally.

Convergent cognition is mostly associated with intelligence such as attentional tasks. Thus, attention restoration is proposed to describe the psychological benefits of human exposure to natural environments [9]. We measured attentional restoration via the Verbal Backward Digit Span Task (Number test), which has been proved to be an effective test for direct-attention performance (a crucial part of short-term working memory) in previous studies and is

normally a verbal task [33, 51–53]. In Number test, a string of numbers starting with three digits was verbally delivered in sequence by the same investigator. Participants were asked to remember those numbers and orally report them in reverse. After each successful trial, the number of digits increased by one for the next trial. Participants had two opportunities for each digit span, and they received 1 point for each digit. For example, the participant who correctly reports five digits at last would receive 5 points.

Divergent tasks are more complex cognitive assessments and relate more to creativity [54]. We chose the validated Guilford's Alternative Uses test (AU test) for creativity evaluation [55]. In AU test, we asked participants to describe as many unconventional uses as possible for an everyday object in two minutes for each environment. Five items were selected including paper towel, plastic bottle, book, umbrella, and mirror which correspond to the five virtual scenes. Two judges evaluated the answers independently based on four criteria. (1) Fluency (the number of relevant and interpretable responses, 1 point each response), (2) flexibility (the number of different categories of responses, 1 point each category), (3) originality (measured by the statistical rarity of the responses in the sample, 1 point for the answer that was given by less than 5% of participants, i.e., the answer that only appeared once), and (4) Elaboration (how detailed the responses are, evaluated on a scale of 0, 1 or 2 points) [55, 56]. Participants received an accumulated score in each test.

Note that there was no baseline measurement done for the cognitive function assessment due to potential learning effect. Adding baseline measurements like those in the physiological indicators would double the number of practices for the participants, thus leading to dramatic increase in learning effect and undermining the reliability of our results.

## Experimental procedure

The classroom setting was kept consistent throughout all visits by arranging the equipment and tables in the same way. Fig 3 illustrates the whole experimental procedure. Upon participants' arrival, they read and signed the consent form. Followed by a brief experiment introduction, investigators would equip participants with the devices (i.e., VR headset and biomonitoring sensors) and gave them safety instructions. Participants then self-oriented with the VR experience and confirmed they do not feel serious discomfort. Then, we explained the approach to perform cognitive tests and gave participants several orientation tests to decrease the learning effect. Specifically, two number tests (with 3 and 4 digits) were given for orientation, and one example of AU test was given to demonstrate the criteria of eligible answers. The orientation was assumed to mitigate the learning effect by giving the participant some chances of practicing so that they would not perform poorly at their first few trials due to fresh contact. After that, participants were exposed to five virtual classroom environments in a random sequence.

In each environment, participants started with a 3-minute rest, i.e., seated in a gray and empty classroom in VR, which allowed their physiological conditions to stabilize and the baseline measurements of physiological indicators to be recorded. Afterwards, participants were exposed to one virtual classroom environment in VR for 3 minutes and allowed to walk and observe the environment freely during the period. Then, without taking off the goggles, the participants were asked to report their current stress level, followed by taking two cognitive tests verbally, which lasted for about 5 minutes. The same procedure was repeated for the other four environments, starting with the 3-minute quiet sitting in the simulated empty classroom which now served as a blanking process to minimize the lingering effect from the previous environment.

After exposure to all five environments, equipment was removed from participants, then they received a 2-minute online check-out survey (S1B File). The questionnaire inquired their

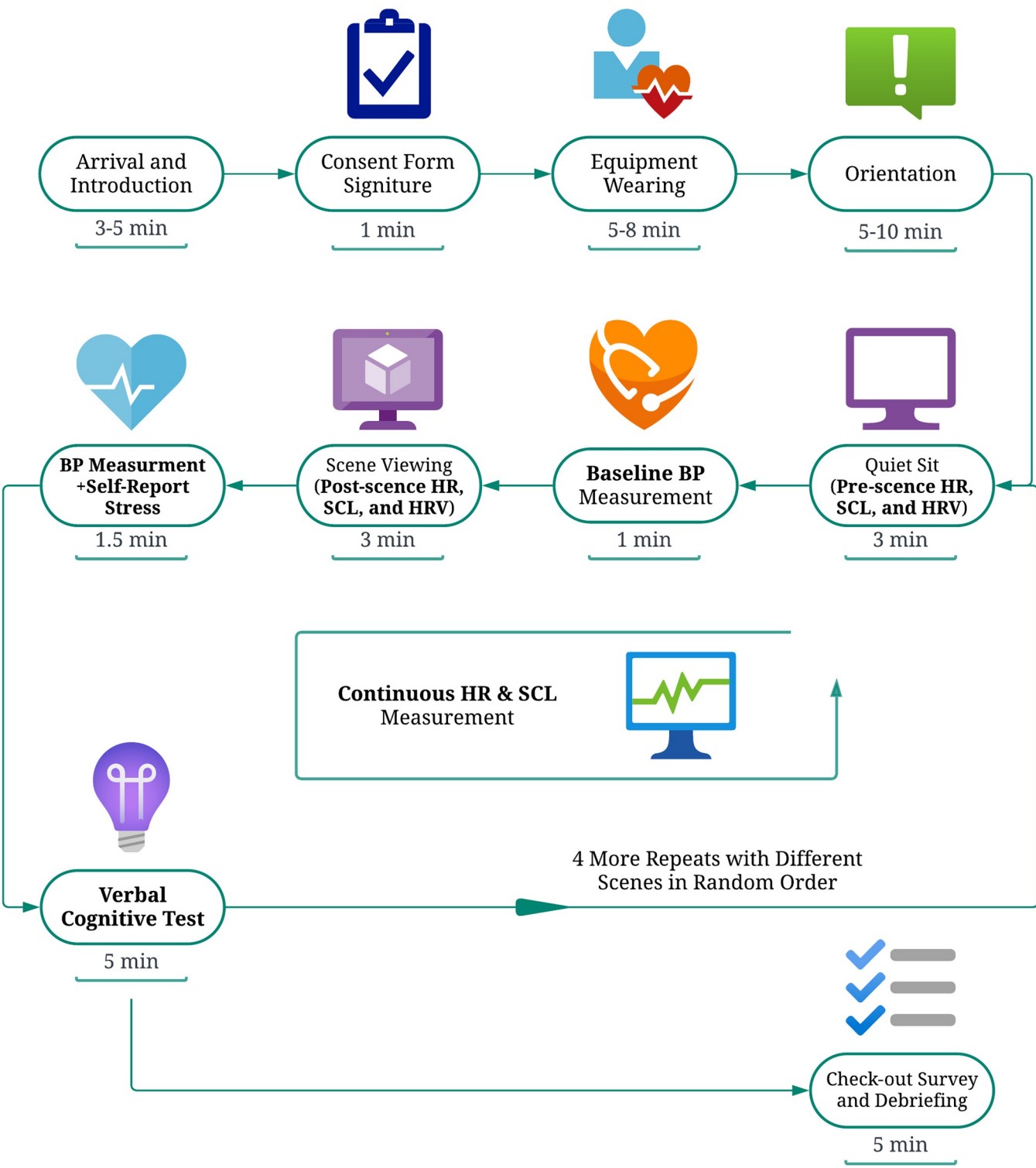

**Fig 3. The flowchart of experimental procedure.** Created with Lucid Visual Collaboration Suite. Time spent in each step is listed under the block, and all the physiological and cognitive assessment data collected are highlighted by bold characters.

feelings of the connection with nature in the virtual classroom scenes (score 1 ~ 10 with no decimal), their preference for the three biophilic patterns (rank 1, 2, 3), their general health conditions, etc. The whole experiment took about 90 minutes individually.

## Statistical analyses

The statistical analyses were performed primarily with R Studio (R version 4.2.1). One-way ANOVA was conducted in Microsoft Excel, and the following post-hoc comparisons were done in Prism Graphpad. A p-value threshold of 0.05 was considered significant for all tests. Fig 4 illustrates the entire procedure of data analysis.

We conducted one-way ANOVA on repeated baseline measures for physiological stress indicators to test whether there were order effects of the five randomized virtual classroom environments. We also examined the potential learning effect in cognitive tests through linear regressions.

The physiological indicators were modeled with linear regressions. Specifically, the measurement right before observing each virtual scene served as a baseline parameter and was deducted from the measurement right after or during the intervention. For BP, the outcome variable was the pre-post scene difference derived from the two momentary BP measures recorded immediately before or after the intervention. For the continuous physiological indicators (i.e., HR, SCL, HRV), we first averaged the measurements over time during the 3-minute intervention and 3-minute blank control respectively, and then calculated the difference which served as the pre-post change (Fig 4). Boxplots were created at this moment to evaluate the potential outliers in the data and visualize the anomaly data points. The data points that deviated far from others (using the boxplot outlier function in R) were deleted to avoid

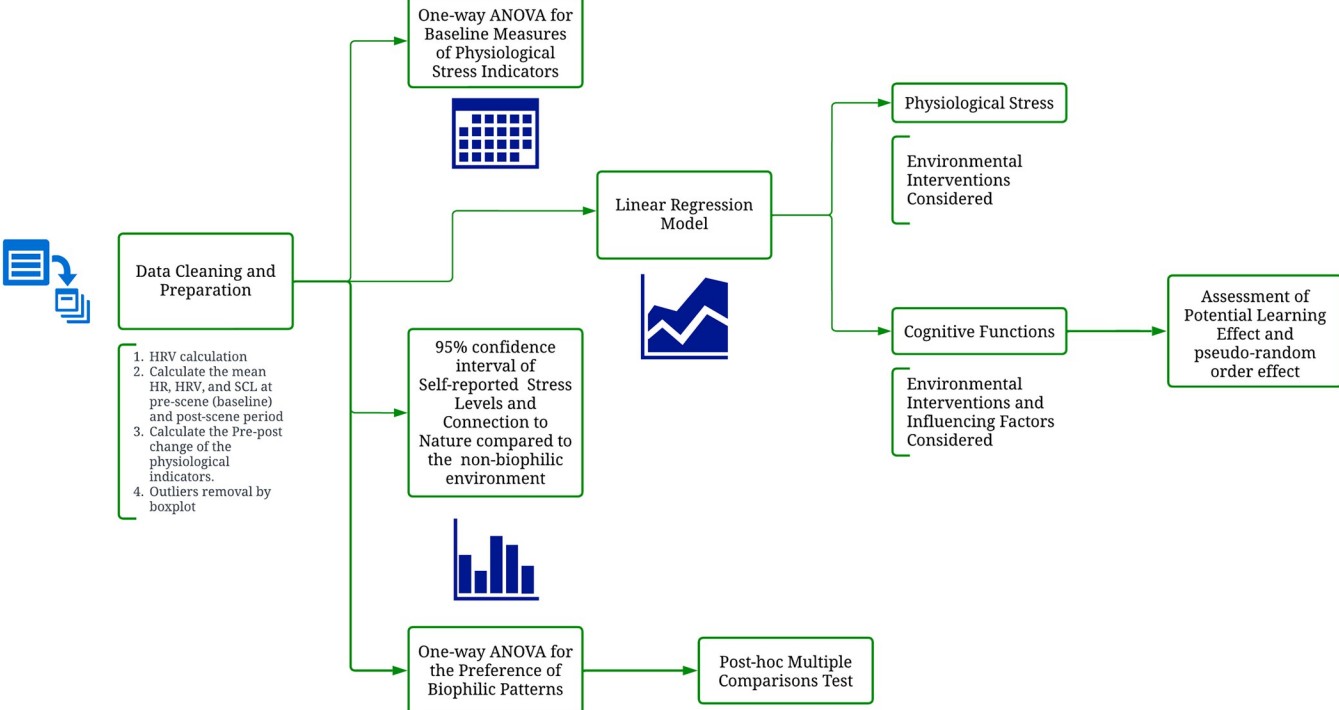

**Fig 4. The flowchart of data analyses.** Created with Lucid Visual Collaboration Suite. All models used are illustrated in corresponding blocks. For clarity, detailed steps for data cleaning are listed below the block instead of creating new branches.

random size effect from accidental factors. After that, linear regression model was built with R package "geepack." We used pre-post change as the outcomes and different biophilic interventions as factor predictors (Model 1)

$$\Delta y_i = \beta_0 + \beta_{1\sim4environment} + e_i \tag{1}$$

where

$\Delta y_i$ is the pre-post difference for BP, or the averaged pre-post difference for SCL, HR, and HRV

$\beta_{1\sim4}$ is the estimated regression coefficient for the four biophilic interventions *environment* is the factors defining different interventions. It is 0 for *Non-biophilic*, 1 for *Indoor Green*, 2 for *Outdoor Green*, 3 for *Turbid Outdoor Green*, 4 for *Combinations*

Effect sizes were then calculated for all the physiological indicators in different interventions. It is expressed as the standardized regression coefficient in this case, which is computed by *lm.beta* function in R. We then used the function *pwr.f2.test* in R to calculate the power with the greatest effect size among all interventions to examine the highest power we can achieve, and the lowest sample size required for this study.

Research showed that cognitive functions are prone to be influenced by general health condition, caffeinated beverages intake, last night's sleep quality, and short-sightedness [57]. Therefore, we constructed a linear regression model for cognitive function performance which took into account those variables. We used post-scene test scores of verbal backward digit span task and the normalized AU test score as outcomes. The predictors included environmental interventions and all the influencing factors (Model 2)

$$y_i = \beta_0 + \beta_{1\sim4environment} + \beta_{5gender} + \beta_{6myopia} + \beta_{7\sim9genhealth} + \beta_{10caffeinated} + \beta_{11}\sim_{14sleepquality} + e_i \tag{2}$$

where

$y_i$ is the test score in verbal backward digit span task, or the z-score of AU test score

$\beta_{1\sim4}$ and $\beta_{1\sim4\ environment}$ have the same meanings as in model (1)

$\beta_{5\sim14}$ is the estimated regression coefficient for gender, short-sightedness, general health condition, caffeinated beverage, and sleep quality

*gender* is 0 for female, and 1 for male

*myopia* is 0 for non-short-sightedness, and 1 for short-sightedness

*genhealth* is 0 for excellent, 1 for very good, 2 for good, and 3 for fair

*caffeinated* is 0 for no caffeine intake, and 1 for otherwise

*sleepquality* is 0 for excellent, 1 for very good, 2 for good, 3 for fair, and 4 for poor

Same as the physiological indicators, the effect sizes were computed.

Lastly, we conducted descriptive statistics analysis for the subjective measurements. For the self-reported stress level (1 ~ 5) and connection to nature (1 ~ 10), the differences between the biophilic interventions and the control scene were analyzed. Regarding the self-reported preference to the three biophilic patterns, besides the mean and standard deviation of the rankings (1, 2, 3), we conducted one-way ANOVA to test the ranking differences of the three biophilic patterns followed by a post-hoc Turkey multiple comparisons test.

## Results

### Demographics and characteristics of the participants and baseline measures

Demographics and characteristics of the 30participants are shown in Table 1. The participants were all undergraduate Chinese student with good self-reported health condition. They had an

**Table 1. Characteristics of study population and baseline physiological measures and other characteristics of visits.**

| Category | *n* (%) or Mean ± SD |
|---|---|
| **Total** | 30 (100) |
| ***Demographics of the 30 participants*** | |
| *Gender* | |
| Male | 14 (46.7) |
| Female | 16 (53.3) |
| *Age* | 19.8 ± 0.9 |
| 18–21 | 30 (100) |
| *Ethnicity* | |
| Asian | 30 (100) |
| *Occupation* | |
| Undergraduate Student | 30 (100) |
| *Self-reported general health condition* | |
| Excellent | 10 (33.3) |
| Very Good | 11 (36.7) |
| Good | 8 (26.7) |
| Fair | 1 (3.3) |
| Poor | 0 (0) |
| ***Characteristics from the 30 visits*** | |
| *Caffeinated beverage drinking* | |
| Yes | 7 (23.3) |
| No | 23 (76.7) |
| *Sleep Quality* | |
| Excellent | 3 (10.0) |
| Very Good | 6 (20.0) |
| Good | 11 (36.7) |
| Fair | 7 (23.3) |
| Poor | 3 (10.0) |

average age of 19.8 (SD: 0.9) years and about a half (16) of them were female. 23.3% (7) of participants reported that they had caffeinated beverage on the day of or the day before the experiment, while two thirds (20) reported that they had relatively good sleep quality the night before the experiment.

No statistically significant differences were observed among the five baseline (pre-exposure) measures for BP, HR, RMSSD and SCL (all p > 0.05) (S1 Table). This suggests that the sequence of the five virtual classroom environments did not affect the physiological responses, and thus it might be the different exposures that accounted for the changes in physiological measurements.

## Stress reactions

The effects of biophilic interventions on physiological stress indicators and self-reported stress levels are shown in Tables 2 and 3 respectively. Data for each participant is provided in S1 Data. No statistically significant pre-post change was found for all the physiological indicators compared to those in the non-biophilic scene. Noticeably, there appeared to be some interesting trends in the data. For example, biophilic intervention 1 to 4 were associated with 1.41 (95% CI: -0.86, 3.68), 1.52 (95% CI: -0.59, 3.63), 1.85 (95% CI: -0.38, 4.08) and 0.33 (95%

**Table 2. Estimated difference (β and 95% confidence intervals) on pre-post physiological changes of stress reaction in the biophilic interventions compared to those in the control non-biophilic scene.**

| Physiological Indicators | Indoor Green | Outdoor Green | Turbid Outdoor Green | Combination |
|---|---|---|---|---|
| ΔSystolic Blood Pressure (mmHg) | 0.25 | -0.33 | 0.89 | 0.74 |
| | (-1.62, 2.12) | (-2.94, 2.28) | (-1.25, 3.03) | (-1.64, 3.12) |
| ΔDiastolic Blood Pressure (mmHg) | 1.41 | 1.52 | 1.85 | 0.33 |
| | (-0.86, 3.68) | (-0.59, 3.63) | (-0.38, 4.08) | (-1.89, 2.55) |
| ΔMean Heart Rate (bpm) | -0.56 | -1.27 | 0.31 | 0.27 |
| | (-2.29, 1.17) | (-3.24, 0.71) | (-0.94, 1.57) | (-1.38, 1.92) |
| ΔRMSSD (ms) | -1.75 | -1.24 | 0.5 | 0.64 |
| | (-6.16, 2.67) | (-5.16, 2.67) | (-2.30, 3.31) | (-3.73, 5.01) |
| Δln(RMSSD) (no unit) | -0.02 | -0.03 | 0.01 | 0.01 |
| | (-0.10, 0.06) | (-0.12, 0.07) | (-0.05, 0.07) | (-0.08, 0.10) |
| ΔMean Skin Conductance Level (mS) | 0.11 | 0.06 | 0.1 | -0.06 |
| | (-0.10, 0.33) | (-0.06, 0.18) | (-0.10, 0.30) | (-0.24, 0.13) |

CI: -1.89, 2.55) mmHg higher diastolic blood pressure (DBP) increases respectively. Neverthe-less, since the results were not significant, it is hard to assess the practical meaning of the trend. Judging from the standardized regression coefficients (Table 4), the effect sizes are rela-tively small using the criteria defined by Cohen et. al. [58] since all of them were smaller than 0.2. We then used the function *pwr.f2.test* in R to calculate the power for the greatest effect size among all conditions (Table 4), which is the ΔDiastolic Blood Pressure in Turbid Outdoor Green Intervention (0.163). Evaluated at significance level of 0.05, u = 4, and v = 25, the power is approximately 0.326. Next, we verified the sample size instead and found that we need at least 79 participants' data to achieve a power of 0.8. This suggested that our study was likely to be underpowered and undermined by small size effects.

Interestingly, there were statistically significant differences in the self-reported stress levels– all the biophilic interventions were related to a decreasing self-reported stress level compared to the control non-biophilic scene. Participants had the lowest stress level in Combination (-0.97; 95% CI: -0.26, 0.12), followed by Indoor Green (-0.73; 95% CI: -1.10, -0.37), Outdoor Green (-0.67; 95% CI: -1.07, -0.26), and Turbid Outdoor Green (-0.47; 95% CI: -0.87, -0.07).

## Cognitive function

Effects of biophilic interventions on participants' cognitive function are depicted in Table 5. The data for each participant is provided in S1 Data. Except for Indoor Green (-0.37; 95% CI: -0.81, 0.08), other biophilic interventions had higher means for the verbal test scores. Specifi-cally, participants could memorize 0.50 (95% CI: -0.02, 1.02), 0.20 (95% CI: -0.24, 0.64), and 0.03 (95% CI: -0.63, 0.70) longer digit spans in Combination, Outdoor Green, and Turbid Out-door Green than in non-biophilic environment, respectively. However, all the changes were not statistically significant at the 95% confidence level.

The effects of all the biophilic interventions were not statistically significant for AU test as well, and there is no obvious trend in the data.

**Table 3. Mean differences in self-reported stress level (95% confidence intervals) between the biophilic interventions and the control non-biophilic scene.**

| | Indoor Green | Outdoor Green | Turbid Outdoor Green | Combination |
|---|---|---|---|---|
| ΔSelf-reported Stress Level (points) | -0.73 | -0.67 | -0.47 | -0.97 |
| | (-1.10, -0.37) | (-1.07, -0.26) | (-0.87, -0.07) | (-1.48, -0.45) |

**Table 4. Effect size expressed as standardized regression coefficient.**

| Parameters | Indoor Green | Outdoor Green | Turbid Outdoor Green | Combination |
|---|---|---|---|---|
| ΔSystolic Blood Pressure | 0.021 | -0.029 | 0.076 | 0.063 |
| ΔDiastolic Blood Pressure | 0.124 | 0.134 | 0.163 | 0.029 |
| ΔMean Heart Rate | -0.048 | -0.109 | 0.027 | 0.024 |
| ΔRMSSD | -0.067 | -0.049 | 0.02 | 0.026 |
| Δln(RMSSD) | -0.033 | -0.054 | 0.023 | 0.027 |
| ΔMean Skin Conductance Level | 0.075 | 0.039 | 0.067 | -0.038 |
| Backward Digit Span Test Score | -0.097 | 0.053 | 0.009 | 0.133 |
| Normalized AU Test | -0.08 | 0.047 | 0.037 | -0.079 |

Note: The regression model is the same as the one outlined in the methods.

Other than the same small effect sizes (Table 4) and low power found in physiological indicators, we also explored other factors causing the low significance. Specifically, we tested learning effect in Number tests (S1A Fig). With the adjustments of other variables, one more test was associated with 0.22 ($R^2 = 0.036$, $p = 0.011$) increase in the digit span, indicating the presence of learning effect. We did not find significant learning effect in AU test ($R^2 = -0.006$, $p = 0.844$) (S1B Fig). Additionally, albeit shuffled by the randomization tool, some environments appeared more frequently at particular positions than others. For example, biophilic-indoor appeared ten times at position A (the first repeat), while it only appeared three times at position C and twice at position D (Fig 5). This pseudo-randomization ordering effect could influence our results negatively.

## Self-reported connection with nature and preference to biophilic patterns

Participants experienced higher levels of connection with nature in Combination (Mean: 6.20, SD: 2.92), followed by Turbid Outdoor Green (Mean: 4.77, SD: 3.18), Outdoor Green (Mean: 3.97, SD: 2.70), and Indoor Green (Mean: 3.03, SD: 2.27) compared to the non-biophilic environment (Table 6).

The descriptive statistics of the self-reported preferences to the three biophilic patterns are shown in Table 7. The one-way ANOVA showed that participants preferred different biophilic patterns in the virtual scenes, i.e., visual connection with nature (e.g. potted plants, windows with trees and sky) gained more preferences than dynamic & diffuse light (i.e., light and shadow) and material connection with nature (e.g. Wooden floor and ceiling) (S2 Table) judging from the post-hoc Turkey comparisons. Notably, 63.3% (19) of participants ranked "visual connection with nature" as their top one choice of preferred biophilic patterns.

## Discussion

We recorded physiological stress indicators, cognitive functions measurements, and self-reported data from 30 participants during exposures to four biophilic interventions and one

**Table 5. Mean differences in Z scores of backward digit span test and AU test (β and 95% confidence intervals) between the biophilic interventions and the control non-biophilic scene.**

| Cognitive Tests | Indoor Green | Outdoor Green | Turbid Outdoor Green | Combination |
|---|---|---|---|---|
| ΔBackward Digit Span Test Score (no unit) | -0.37 | 0.2 | 0.03 | 0.5 |
| | (-0.81, 0.08) | (-0.24, 0.64) | (-0.63, 0.70) | (-0.02, 1.02) |
| ΔNormalized AU Test (Z-score, no unit) | -0.2 | 0.12 | 0.09 | -0.2 |
| | (-0.53, 0.13) | (-0.19, 0.42) | (-0.22, 0.40) | (-0.51, 0.11) |

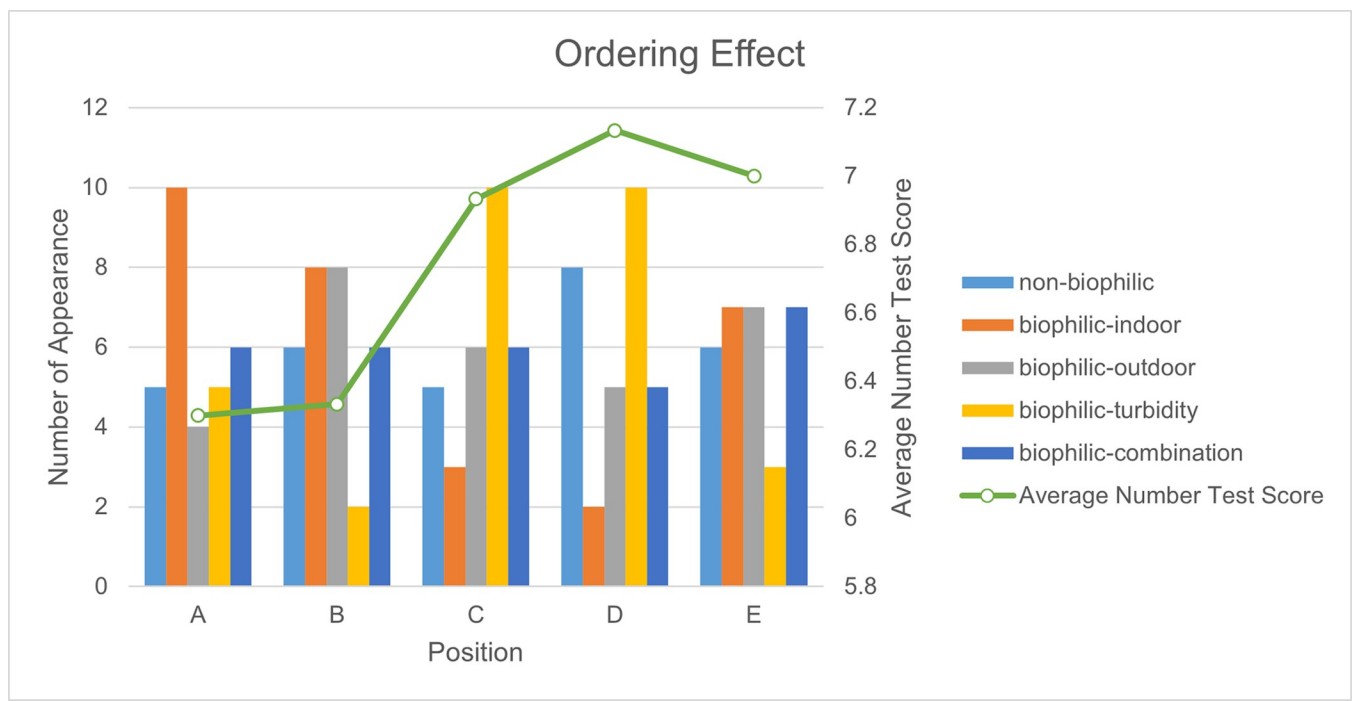

**Fig 5. Ordering effects in cognitive tests.** A-E stands for the appearance order of the environments. A means the environment that appears first in one complete experiment, B means the second environment, etc. Bar charts represent the appearance numbers of each intervention at the position (A-E). "Number of Appearance" measures how many times the environment is ordered at this position among all the experiments. The green line is "Average Number Test Score" which is the mean score for all the number tests done after each scene at the position. For example, the dot at position A stands for the average score for all the number tests done after the first environment.

control non-biophilic scene. After removing the anomalies, no significant differences were found for physical and cognitive measurements. However, the estimated coefficients for subjective stress ratings suggested stress relief effects in all the interventions. Thus, there existed a discrepancy between participants' objective stress measurements and subjective feelings. Meanwhile, participants' attention and creativity showed no consistent improvement across the interventions. All the results were not statistically significant, suggesting potential random effects. We also believe that the study is underpowered due to small sample size and small effect sizes. In general, participants subjectively felt significantly less stress and more connection with nature in all the interventions. Therefore, we were able to partly confirm our hypothesis about the positive but different impacts of the biophilic interventions from the mental side, but could not confirm the negative impacts of the turbidity based on the current evidence.

## Stress reaction

None of biophilic interventions showed significant improvement in objective physiological indicators. This is likely due to the small effect sizes, which in turn led to a small power for our

**Table 6. Mean differences in self-reported connection with nature (95% confidence intervals) between the biophilic interventions and the control non-biophilic scene.**

|  | Indoor Green | Outdoor Green | Turbid Outdoor Green | Combination |
|---|---|---|---|---|
| ΔSelf-reported Connection with Nature (points) | 3.03 | 3.97 | 4.77 | 6.2 |
|  | (2.19, 3.88) | (2.96, 4.97) | (3.58, 5.95) | (5.11, 7.29) |

**Table 7. Mean of preference (rank 1, 2, 3) to the three biophilic patterns (standard deviation) in the biophilic interventions.**

|  | Visual connection with nature | Dynamic & diffuse light | Material connection with nature |
|---|---|---|---|
| Preference to biophilic patterns | 1.47 | 2.07 | 2.47 |
|  | (0.68) | (0.78) | (0.68) |

study. All the effect sizes were smaller than 0.2 (Table 4), indicating a higher requirement for sample size. A minimum of 79 participants was necessary to achieve a power of 0.8 if interrogated with the largest effect size among all interventions (0.163).

However, there were two discrepancies worth discussing. Firstly, some interventions had similar physiological changing patterns, though the results were not significant. Indoor Green and Outdoor Green induced similar changes in stress indicators except for SBP, while Turbid Outdoor Green and Combination had similar measurement changes except for SCL. Research shows that people's perceived restorativeness of natural environments can be influenced by their feelings of connectedness to nature [59]. In our study, participants reported that they experienced more "light" in Turbid Outdoor Green, making them feel more connected to nature. Therefore, we suspected that similar physiological responses were related to subjective feelings of nature. This is consistent with the self-reported connection to nature where Indoor Green and Outdoor green received lower score than Turbid Outdoor Green and Combination.

Secondly, participants self-reported significantly lower stress levels after exposure to biophilic interventions, which is contradictory to the non-significant changes in the physiological measurements. This also happened in Verzwyvelt et al.'s study on oncology patients who reported more willingness to contact nature, but no significant stress relief responses were found physiologically [60]. Additionally, Emamjomeh et al. found significant decline of negative mood in in situ and IVE biophilic environments compared to the control group, while no alleviation of physiological stress responses was observed (mean HR, Baevsky's stress index, SD2 (%), mean RR (ms), RMSSD (ms), and SD1 (%)) [61]. Since there was no statistically significant effect found, we cannot ascertain whether this is due to a real ignorable influence on physiological indicators from the biophilic interventions or a direct impact from low powers. This potential discrepancy suggests that although participants in virtual reality might not experience physical stress relief, their perceived stress in biophilic simulations could be released. Studies revealed that intrapersonal emotional competences (the ability to recognize and interpret one's own emotions) are important for perceived stress and certain aspects could help people avoid stress [62, 63]. Therefore, our findings have implications that incorporating biophilic elements in virtual classrooms might be an effective stress management intervention for students.

Another interesting finding is the consistently higher BP means across all biophilic interventions. This increase, although not statistically significant, may be caused by the absence of stress induction during the experiment. It also explains the contradiction with previous studies which found restorative effects of biophilic environments on BP through immersive video either shown on TV [64] or on VR simulations [30, 65]. This result suggested potential arousal from biophilic interventions. Interestingly, participants implied boredom in the non-biophilic environment in the immediate feedback after the experiment. Therefore, this sympathetic arousal might be caused by the novelty of natural elements in biophilic interventions.

## Attention and creativity

Although not significant, three out of the four interventions are associated with higher mean values for scores, which is consistent with the ART [11]. Previous research showed that

simulated and actual natural environments would enhance the attention-required ability [66]. Among all virtual environments, Combination was associated with the highest improvement in the number test, while participants performed worst in Indoor Green. Resembling the results in the stress reaction, this difference observed between the virtual environments might result from subjective feelings of connection with nature. Since participants felt most connected to nature in Combination, their involuntary attention was likely to be directed, freeing the voluntary mind volume [11]. The potential learning effect in the number test is one possible factor leading to insignificance in the statistical analysis. This is one unexpected result because we randomized the order of the virtual environments the participants experienced, which should have counteracted the learning effect. To further explore the possible cause of insignificance, we counted the appearance number of each environment at each position. The results suggested a pseudo-randomization effect because certainly some environments appeared more often than others at particular positions (Fig 5).

However, it is still hard to explain the learning effect by considering the unbalanced order. Judging from Fig 5, we could intuitively attribute the increase in number test score to the appearance of biophilic-turbidity environment and the low score to appearance of biophilic-indoor environment. The rule would follow until it comes to position E. E has more biophilic-indoor and less biophilic-turbidity appearance than C, but its average test score was higher. Meanwhile, if indeed the number test performance was associated with certain environments, we should observe significant linear relationships in previous statistical analysis, while there were no such results present. Therefore, it is possible that the insignificant results produced from this study were caused by a complex interactive effect from both the learning effect and the unbalanced order of environment sequences. Besides, the standardized regression coefficients also witnessed small effect sizes for cognitive assessments (Table 4).

Meanwhile, no significant improvement of creativity was observed, and thus the conducted study was unable to confirm Yin et al.'s results that participants revealed significantly more creativity and less attention [30]. Still, since the effect sizes were small and our study is possibly underpowered, we cannot make any conclusion about whether the simulated biophilic environments will have any effect in divergent task.

## Biophilic design in the Metaverse

Going against the inclination that reproduces the physical world in the virtual world and uses virtual environments to predict human responses to the actual natural elements [33, 61, 67, 68], this study designed biophilic environments for the virtual world itself and explored people's responses to the virtual biophilic elements. In this study, a single-user system for biophilic design was adopted where only one participant was in the virtual environment at a time, and no research with a multi-user system has been found according to a recent critical review [11]. However, the Metaverse is proposed to be a "more embodied Internet" where each user is represented by an avatar, and interactions exist among avatars in a shared social place [1]. This suggests that a multi-user system for biophilic design in the Metaverse needs further investigations.

## Strengths and limitations

This study is among the few studies on biophilic design in a virtually built environment and has several strengths. Firstly, we framed the study in the context of Metaverse by investigating the health effects of biophilic design in the virtual world, where people would probably spend most of their time in the future. Secondly, the verbal cognitive tests minimized the potential visual interferences caused by questions showing on the virtual screen in the VR goggle.

Additionally, the insignificant changes in the physiological stress indicators and the significant drop in the self-reported stress levels indicates an interesting gap between the objective assessments and subjective feelings of one's stress. Fourthly, the study used a head-mounted display (HMD) with a three-dimensional environment, offering a more immersive experience for participants than two-dimensional pictures [50, 69]. Fifthly, we largely made sure that participants did not have serious cybersickness during the experiment. To achieve that, we offered an orientation for participants to get comfortable with the VR goggle and ensured immediate stop of the experiment if they feel any discomfort at any time. This helped prevent the potential negative effects on participants' stress level and task performance in the virtual environments [70, 71]. What is more, randomizing the sequence of exposures to the virtual environments could help reduce order effects and the potential bias when participants compare different environments. Lastly, the repeated physiological and cognitive measurements on the same individual could help control potential time-invariant factors such as gender and socioeconomic factors.

The study also has some limitations. Firstly, given the small effect sizes (Table 4) and low powers, the sample size should be larger to detect any significant change in participants physiological stress reactions and cognitive functions. Secondly, due to the limitations of the computer configuration, the virtual environments became choppy in the VR viewer for a few times, which could possibly affect participants' perceptions in virtual environments and in turn, influence their stress reactions and performances in cognitive tests. Thirdly, the experiments were conducted all through the day. As a result, the mental condition could vary among participants (i.e., different people tend to be more unfocused at different time of a day) and thus the effect of differences in the visit time may not be negligible. Some other possible intervening variables affecting the physiological measures that were not incorporated in the analysis include smoking, hunger, and gender differences. Fourthly, we delivered both cognitive tests verbally to avoid the potential visual interruption caused by the switch of interface in VR goggle. Although the Backward Digit Span Task (Number test) by convention is delivered verbally, the feasibility and reproducibility of verbal AU test have not been substantially validated. Finally, to minimize the potential learning effects, we did not perform baseline measures for cognitive tests. We still detected a learning effect in the number test but not in the AU test. However, this learning effect could partly be offset by the randomization of the virtual environments.

There are several recommendations for further metaverse-related environmental design study. Besides the randomization, immersive experience, and larger sample size, the simulated environment should have high quality that would put participants in a continuous vivid setting. In terms of immersive experience, studies found that interactivity might play a significant role in the restorative impact of natural environments in VR, and engaging in playful interactions could help managing short-term stress [72]. Additional control of confounding variables like time of experiments and combination of different orders of exposed environments should also be implemented.

## Conclusions

In this randomized crossover study, we exposed 30 university students to various virtual biophilic classroom scenes using VR to gauge the influence of visual interactions with nature on their stress levels and cognitive performance. To enhance the participants' engagement with the virtual setting, all cognitive tests were administered verbally. Despite observing small effect sizes and the influence of pseudo-randomization, there were no significant or consistent shifts in physiological stress responses and cognitive abilities. However, participants did report feeling notably less stressed and felt a deeper connection to nature when in biophilic settings,

including those with turbidity. Our findings add a unique dimension to existing literature by focusing on university settings and student demographics, with suggestive evidence that the potential advantages of integrating biophilic design elements into the Metaverse and real-world university classrooms to enhance stress outcomes. To delve deeper into the effects of biophilia within the Metaverse, future studies should prioritize refining students' perceptions of biophilic components and their immersive experiences in the digital realm.

## Supporting information

**S1 File.** (A) Screening survey for participant recruitment which introduces the study, collects contact information, gender, age, ethnicity, and relevant health conditions. (B) Check-out survey which collects general health conditions, perceived stress levels, feelings of the connection with nature in the virtual classroom scenes, and preference for the three biophilic patterns. (DOCX)

**S1 Data. Source data for each subject.** This includes raw measurements or those after the primary computation of three physiological stress indicators, two cognitive tests, self-reported stress level, connection with nature, and preferences for different biophilic patterns of each participant.
(XLSX)

**S1 Fig. Learning effects in cognitive tests.** (A) Learning Effects in Verbal Backward Digit Span Task (Number test). With the adjustments of other variables, one more test was associated with 0.22 (95% CI: 0.051, 0.389) increase in the digit span. (B) Learning Effects in Alternative Use Test (AU test). No significant learning effect was detected in AU test.
(ZIP)

**S1 Table. ANOVA of baseline measures of physiological stress indicators.**
(XLSX)

**S2 Table. One-way ANOVA (sheet 1) and post-hoc comparison (Tukey's multiple comparisons test) of preferences for different biophilic patterns (sheet 2).**
(XLSX)

## Acknowledgments

We extend our appreciation to express our gratitude to Keping Wu, Baozhen Luo, and Chi Zhang from the Center for the Study of Contemporary China at Duke Kunshan University for pioneering seed initiatives that catalyzed this research project. Our gratitude also goes out to Zerui Tian, Yu Leng, Junyi Li, and Lihan Huang for their research assistance in preliminary tests, data collection, and processing. Finally, our heartfelt thanks to all the student volunteers from DKU who took part in this study.

## Author Contributions

**Conceptualization:** Xinyi Wen, Jie Yin, John S. Ji.

**Data curation:** Jicheng You, Xinyi Wen, John S. Ji.

**Formal analysis:** Jicheng You, Xinyi Wen, Linxin Liu, John S. Ji.

**Funding acquisition:** Jicheng You, Xinyi Wen, John S. Ji.

**Investigation:** Jicheng You, Xinyi Wen, Linxin Liu, John S. Ji.

**Methodology:** Linxin Liu, Jie Yin, John S. Ji.

**Project administration:** Linxin Liu, John S. Ji.

**Resources:** John S. Ji.

**Software:** Linxin Liu, John S. Ji.

**Supervision:** Linxin Liu, Jie Yin, John S. Ji.

**Validation:** Jicheng You, Xinyi Wen, Linxin Liu, Jie Yin, John S. Ji.

**Visualization:** Jicheng You, Xinyi Wen, John S. Ji.

**Writing – original draft:** Jicheng You, Xinyi Wen.

**Writing – review & editing:** Jicheng You, Xinyi Wen, Linxin Liu, Jie Yin, John S. Ji.

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
