## [Decision Letter · Decision Letter 0]

23 Feb 2023

PONE-D-22-35023Biophilic classroom environments on stress and cognitive performance: A randomized crossover study in virtual reality metaversePLOS ONE

Dear Dr. Ji,

Thank you for submitting your manuscript to PLOS ONE. After careful consideration, we feel that it has merit but does not fully meet PLOS ONE’s publication criteria as it currently stands. Therefore, we invite you to submit a revised version of the manuscript that addresses the points raised during the review process.

 Two Reviewers have evaluated the manuscript and both found notable weaknesses and minor aspects that deserve Authors' attention. I agree that the manuscript deserves to be notably improved especially in terms of adherence to previous literature, clarity in methodology and presentation of results. 

We look forward to receiving your revised manuscript.

Kind regards,

Stefano Triberti, Ph.D.

Academic Editor

PLOS ONE

Journal Requirements:

3. Please amend your current ethics statement to address the following concerns:

a) Did participants provide their written or verbal informed consent to participate in this study?

“we want to thank the Center for the Study of Contemporary China (CSCC) at Duke Kunshan University (DKU) for providing generous funding.”

“J.S.J. received funding from the Center for the Study of Contemporary China (CSCC) at Duke Kunshan University for an Undergraduate Research Grant for the academic year 2020-2021 (https://www.dukekunshan.edu.cn/cscc/). The funder had no role in study design, data collection, and analysis, decision to publish, or preparation of the manuscript.”

Reviewers' comments:

Reviewer's Responses to Questions

**Comments to the Author**

1. Is the manuscript technically sound, and do the data support the conclusions?

Reviewer #1: Yes

Reviewer #2: No

2. Has the statistical analysis been performed appropriately and rigorously? 

Reviewer #1: Yes

Reviewer #2: No

3. Have the authors made all data underlying the findings in their manuscript fully available?

Reviewer #1: Yes

Reviewer #2: Yes

4. Is the manuscript presented in an intelligible fashion and written in standard English?

Reviewer #1: Yes

Reviewer #2: No

5. Review Comments to the Author

Reviewer #1: Experimental design and procedure were methodologically correct and appropriate. The conclusions are in line with observed data and statistical analysis have been performed appropriately. All necessary matherials are available, including data. Overall the manuscript is intelligeble and written in standard English. Additional comments were reported in the attached file.

Reviewer #2: In this study, the authors aimed to assess the effects of biophilic virtual environments on stress levels and cognitive performance in 30 university students. To do so, the students were immersed in 4 different biophilic virtual environments and 1 control virtual environment using a VR headset. Before, after and during the use of the VR headset, the participants performed different test and answered multiple questions. The study mainly revealed null effects.

Although the topic is relevant and interesting to a broad readership, I believe that the study design, unfortunately, was not adequate to address the authors’ potential research question and hypotheses. I believe that the study is underpowered and the results are not described clearly enough to support the conclusions. In addition, the manuscript fails to address how the findings relate to previous research in this area. The authors should rewrite their Introduction and Discussion to reference the related literature, and the results section to sufficiently describe the conducted analyses. Throughout the manuscript, but especially in the discussion section, the authors describe hypotheses, prior findings and their own findings intuitively rather than comparing it to other literature or providing evidence from prior research. Also, when findings are compared to literature it is mostly to the same one paper (ref 31).

Moreover, the manuscript would also benefit from a proofreading by a native English speaker to improve readability and take out some language mistakes and unclear sentences.

Major issues

1. In general, the introduction is well-structured, but I believe that the purpose of the study, in light of prior studies, is not clear. How does the study relate to other studies, how is it different? What gap in literature do the authors address with their study? And how does that gap lead to the research question and potential hypotheses?

2. The hypothesis is quite vague. What do the authors mean with different levels of physiological stress reduction and cognitive function improvement? Do the authors expect less stress reduction in some conditions and more in others? Please specify.

3. Thee research question stated in the abstract does not fully align with the hypothesis mentioned in the introduction section. How do they relate to each other?

4. Fig 3 is should be adjusted to avoid confusion about the specific study design and procedures. The figure is not easy to read. Please write it out in full in the text and keep only the larger procedural steps in the figure, eg, adding which outcome measures are assessed at which time point and visualization of the different steps a participant takes, would make the procedure more clear.

5. I believe that the study is underpowered. The authors perform a lot of statistical tests and have a small sample size. The authors should perform a power analysis and, if needed, also mention the power as a limitation. Also, ss the hypotheses are not clearly stated and the study was not preregistered, it is not clear which statistical tests were planned and which one were exploratory.

6. The description of the study design does not clarify how this is a crossover study as there are no clear cohorts. Also, between different VR environments there was hardly a wash-out period. How can the authors be sure that the effects in the second, third, fourth or fifth environment are not lingering effects from the first environment? How much physiological effects/stress relief can you still expect after multiple environments? I believe that this is a design choice that severly impacts their results, and this should be mentioned as a limitation.

7. The authors have not mentioned all the necessary statistics, eg., R2 and p-values in the results section. Also, when reporting the results of a one-way ANOVA, authors need to describe the independent and dependent variables, the overall F-value and the corresponding p-value, and also the results of post-hoc comparisons. This information is missing in the manuscript, but the authors do conclude and describe that there are differences between the VR environments.

8. Line 185: I believe that crucial information is missing in the description of the physiological sensors used in order to be able to replicate the study. First, information of, for example, sampling rate for HRV is missing, were artifacts removed, if so, how? How was the data preprocessed, how was the data calculated to use in analyses?

9. Line 267: what type of ANOVA, which factors (with which levels) were included in the analyses? Also, did the authors check the assumptions? Which factors were included in the linear regression analyses?

10. Line 270-272: “In all the analyses of physiological indicators, the measurement before each virtual scene observation served as a baseline parameter that is deducted from the measurement after or during the environmental exposure.” This information is too vague. What do the authors mean with the measurement before? Do the authors work with an interval, did they take an average of that interval? What is considered before, how much time? What do they mean with ‘the measurement ‘after or during the exposure’? How was the physiological data processed before analyses?

11. Line 326: It is not clear what the authors mean with five baseline measures. Were there 5 cohorts of specific orders of the VR environments? Also, are these the true baseline measures, namely the outcomes measures assessed for the very first time, before any VR environment was experienced, or do these baseline measure refer to the average of each outcome measure before each new VR environment?

12. Line 393: The authors report the means and SD of the preference scores, but do not report statistical results to support the statement that participants experience higher levels of connections in certain environments. If there are no significant differences between environments, the authors should state this. Also, as the description of the ANOVA analysis uses different wording than in the table, it is not clear which environments are preferred over others.

13. Line 413: The conclusion drawn by the authors is not supported by the results. In the result section, the authors have stated that there were no significant difference and now they suggest ‘partial stress relief’. This does not seem correct to me, nor is it clear what the authors mean precisely. Also, what is the suggested discrepancy between objective stress measurements and subjective feelings?

14. It is confusing for readers that the authors first describe results, and then state ‘but these results were not statistically significant’. When the results are not statistically significant, the authors should not describe them in that manner.

15. Line 420: The authors state that ‘results were not statistically significant, suggesting small effect sizes’. This is not a correct statement. The authors can calculate effect sizes and report them to make an evidence-based statement.

16. Figure; the results are not clearly shown in the figure. It is quite a busy figure, showing a lot of information, but it is not clear what ‘position’ refers to, nor what ‘number of appearance or average number test refers to.

17. Line 480; The authors make incorrect claims about the attention improvement, as they stated prior that there was a learning effect. (line 388)

18. Line 490: With this statement the authors show that their study, unfortunately, was not ideally designed for their research questions. I believe that the learning effect is independent of the order of the VR environments. The learning effect occurs due to repeated testing, up to five times. You can expect that even after 1 cognitive test participants will perform better the next time, and here the authors have repeated it 5 times in 90 minutes due the pre and post assessment sessions (if I have understood the study correctly). Authors should include information on test-retest reliability and repeated assessment. It was also not clear to me, until line 562, that there was no baseline cognitive measure. The authors should clarify this in the study design figure.

19. In the ‘strengths and limitations section’ the authors make some statements without backing them with evidence. For example, as a 3rd strength the authors state that the comparison between physiology and self-reported stress levels is a strengths, but the authors have not formally compared this (ie. with correlation analysis), as they only described both results. Also, the authors state that they provided a very immersive experience, but as they did not formally assess sense of presence/immersion (ie with IPQ), this is again an unfunded statement.

20. The statistics are not reported correctly, which makes the results section a bit confusing to read. The authors on multiple occasions start describing the results to then state that the results were not significant. This is confusing for the reader. The conclusions are not supported by the results, the authors describe results as if they were significant and the discussion section would benefit from a clear focus. Also, the authors report multiple findings in the discussion section that were not described in the results section (eg line 492-497: order of the presented environments).

21. Line 413: The authors mention new information that was not mentioned in the methods section nor results section, namely the removal of anomalies in the data. The authors should include this analysis step in the methods and analysis section and clarify what is considered as an anomaly.

22. Line 444: Seen as there are no significant findings concerning the physiological outcome measures, the authors should not interpret the results as if there were significant differences between these groups.

Minor issues

23. Considering that the primary outcome measures of the study are the physiological measures (though not stated explicitly), the authors should consider including more information in the introduction section on how physiological measures are linked to stress, as this information is currently missing.

24. Line 135 and 137. It is not clear what references 35 and 31 actually refer to, or what evidence they provide. Please clarify this in the text. Describe the 3 biophilic design patterns and explain why they are suitable for short exposure time, and what is considered a short exposure time. Perhaps it might even be better to already explain this in the introduction section.

25. Line 201: Stress level rating, what did the 1 refer to, what did the 5 refer to? W

26. Line 214: Cognitive tests delivered verbally might be a limitation as they were not validated as such. It is not clear to me how the tests were delivered. If the participants were still seeing the VR environment, doesn’t this create some sort of sensory overload?

27. Line 227: The authors state that ‘divergent tasks are more complex cognitive assessments and relate to more creativity’, without providing a reference for this claim. Please do so.

28. If participants are allowed to move during the physiological assessments (during VR use), how did you correct for movement artefacts?

29. Line 261: Did the online check-out survey consist of validated questionnaires? Did the authors use open-ended or close-ended questions?

30. Line 269: how was the order of the randomized virtual environments determined? Were there groups of the same order? The authors should provide information on how many times each environment was experienced in which place in the order? If this was completely random and, for example, environment 1 was only 2 times in the second place, whereas environment 3 was 20 times in the second place, then how can you assess potential order effects? As this (new) information is later addressed in the discussion section, it should be addressed first in the methods and results section.

31. The statistical analysis section does not describe which statistical program is used, what the level of significance is, which post-hoc corrections were applied. Also, no information on checking of assumptions is provided.

32. The methods section would benefit from a separate ‘outcome measures’ section detailing the different outcome measures and the time points at which they are assessed. The authors could do this visually in the procedure figure.

33. The results section mentions both diastolic and systolic blood pressure, whereas the difference between both or relevance of assessing both is not explained in the methods section. Please do so.

34. Stress reactions section: For clarity, the authors should state upfront that the results were not statistically significant instead of in the end. The authors should do this throughout the results section and discussion section.

35. Table 3: How was the self-reported stress level per group calculated? As these are negative numbers, are these difference scores? If so, please mention this in the results section, as well as in the methods section.

36. Line 362: Was there a significant difference in average (pre-to-post changes in) stress levels? Or is this a descriptive analysis?

37. Line 435: I do not agree with the authors that the ‘Outdoor Green’ is an open space. It is still a classroom, so rather considered as a closed space. The authors should add literature on what is considered open and closed space and how this feature relates to their findings, if the authors want to mention this feature. The mentioned reference does not support the statement, or is not good comparison material, seen as in that study there was a noticeable difference in the scale of the space shown. In this study, the scale of the virtual rooms is the same, as it is the same room.

38. Line 440: Can the authors clarify what they mean with ‘subjective feelings of nature’ and change it accordingly in the manuscript?

39. Lines 518-519: The authors should elaborate on the why and how of this statement. What do they refer to with their reference?

Additional minor remarks

40. Line 36, 77: Grammatically incorrect sentence. I believe that the word ‘that’ is missing.

41. Line 54: decreased instead of decreasing

42. Line 76: Please rephrase. This sentence is quite a bold statement that assumes causation based on completely unrelated studies.

43. Line 116: ref 34 is not a correct reference for the statement that is made. The authors refer to another article of one of the authors in which they state that using a cross-over design would provide a larger power with the same number of participants, but that study did not actually assess this. Please use a correct reference to indicate your choice for this design. Also, was a power calculation conducted beforehand? If not, please provide a power calculation in the manuscript and , if the power is insufficient, add this information in the limitations section.

44. Line 126 should be inclusion criteria, not including criteria.

45. Line 157: should state NVIDIA instead of NVIDA

46. Line 169: should state refresh rate instead of fresh rate.

47. Line 176: What do you mean with ‘obtained by calculation’, can you please specify?

48. Line 210: I do not fully understand this sentence. Do the authors state that VR has the higher potential to boost cognitive function than traditional methods or does VR provide a better method to assess cognitive function that traditional methods? The wording is not clear on this. Also, the study that is referred to is quite old in terms of VR research. I believe there is more recent work that can provide some background on using VR for cognitive assessment. Also, if referring to VR for cognitive assessment, why do the authors then choose verbal tests? Are the tests that were used validated to use them in this manner? If there is no VR assessment of cognitive function in the study, there is no need to provide background information on the matter.

49. Instead of Characteristics from the 30 visit, which is confusing wording, the authors might want to use ‘circumstantial factors’ or something similar.

50. Why are the participants divided into two age groups; 18-19 and 20-21? Please provide mean age and SD in the table, as was described in the text.

6. PLOS authors have the option to publish the peer review history of their article (what does this mean?). If published, this will include your full peer review and any attached files.

Reviewer #1: No

Reviewer #2: No

---

## [Author Response · Author response to Decision Letter 0]

26 Apr 2023

Reviewer 1

The current research investigated the effects of a virtual indoor environment with natural elements on cognitive functions and stress levels, based on the biophilia hypothesis. The authors conducted a randomized crossover study, proposing 30 Chinese students to four virtual classroom environments varying in biophilic elements and turbidity. They measured physiological indicators of stress (blood pressure, heart rate variability, and skin conductance level), as well as self-report measures of stress and connection with nature, and conducted attention and creativity tasks to assess cognitive functioning. Even though they did not find significant changes in stress reactions or cognitive functions, subjects showed a reduction in self-reported stress, and positive connections with nature were observed. The authors concluded that the addition of biophilic elements in the virtual environments could be beneficial to promote well-being. 

This is very interesting research investigating possible new ways to exploit technological opportunities to promote human health. Some adjustments and clarifications might improve the work. I suggest authors consider the following comments and suggestions:

1. Introduction: It is not clear why you talked about “virtual reality Metaverse” instead of mere virtual reality. You might better explain this expression in the introduction;

Thank you for the suggestion. Previously, we talked about “virtual reality Metaverse” because we wanted to set the context of the study as the emerging Metaverse. However, we revised it to “virtual reality” as we now discovered that it is confusing to term it “virtual reality Metaverse.” In addition, we did not use VR to bring participants to the Metaverse, i.e., the digital parallel universe connecting to the real world.

2. Introduction: you might better present the current state-of-the-art about the use of virtual natural environments to promote well-being, focusing on a psychological perspective (e.g., Simone Grassini. 2022. The use of VR natural environments for the reduction of stress: an overview on current research and future prospective. In Proceedings of the 33rd European Conference on Cognitive Ergonomics (ECCE '22). Association for Computing Machinery, New York, NY, USA, Article 4, 1–5. https://doi.org/10.1145/3552327.3552336)

Thanks for providing this wonderful literature. We did not realize the missing of such information and now we added the current state-of-the-art about the use of virtual natural environments to promote well-being in the Introduction section (line 77-80).

3. Pag. 5 lines 101-102: “The guiding role of visual senses in creating a perception” and “objective virtual exposure assessment” sound like abstract and unclear concepts. You might better explain these expressions (i.e., Do you refers to the dominance of the visual system in perception?). 

We rephrased the expressions and added some examples to explain this. “Combining VR, eye-tracking, and wearable biomonitoring sensors, their studies supported the dominance of visual senses in creating perceptions and provided a potential tool for objective virtual exposure assessment (e.g., measuring physiological stress reactions such as blood pressure and heart rate).” (line 83-87).

4. Pag. 6 line 105: “short-term health” expression is not clear: what do you mean by short-term?

By “short-term health,” we mean immediate physiological and cognitive responses (i.e., physiological & perceived stress levels and cognitive tests performance) to the virtual scenes during the 90-minute experiment session (line 92-96). We did not assess the participants’ health condition after the experiment.

5. Pag. 7 Environment Simulation section: it is not completely clear why you choose to use indoor virtual environments since it is not fully exploited VR opportunities; I expected to find immersive naturalistic virtual environments instead of a classroom replication with some plants. Explain better your choice. You might also expand this aspect in the discussion/ future works. 

Thanks for raising this confusion! The main purpose of this study is not to fully exploit VR opportunities, so we did not choose the naturalistic virtual environments. We wanted to investigate the short-term health responses in virtually built environment with biophilic elements; and we used VR due to the emerging Metaverse and the recent emerging use of health-related VR systems. Detailed explanation can be found in the Introduction section, especially in the first and second paragraph (line 45-71). 

6. Pag. 7 Study Population: were participants instructed to avoid smoking the day of the experiment or check for other possible intervening variables affecting physiological measures (excluding caffeine and subjective report of sleep quality)?

Participants were not instructed to avoid smoking. This is because participants come from a campus that is mostly smoking-free. However, we can add this to future study. The investigated possible intervening variables are listed in Statistical Analysis section (line 327-346).

7. Pag. 10 line 206: did you use a VAS for the stress level rating? What was the item? 

Thank you for raising this concern! We did not use a VAS or other empirical test forms for the stress rating. We let the participants verbally rate their stress level from 1 to 5 (no decimal) – 1 refers to “not stressful” and 5 denotes “extremely stressful” (line 205-206). Reason for such rating system is that our experimental setting requires continuous mounting of the goggle and minimal movement of the physical sensors. This extremely limited the complexity of the rating system we can use.

8. Pag. 11 lines 217- 227: Why did you choose attention and creativity? Are there previous studies specifically focused on these domains? You might better argue your choice. Additionally, in line 210 you stated the potential benefits of VR in boosting cognitive function assessment, but your paradigm tried to manipulate cognitive functions instead of assessing them. 

We chose attention and creativity because (1) they can be assessed through validated cognitive tests, (2) Yin et al.’s studies found that they could potentially be influenced by short-term environmental exposures in VR. We added this explanation in the manuscript (line 214-217).

Secondly, please kindly refer to the response to Reviewer 2’s comment #48 about using VR for cognitive function assessment.

9. Pag. 11 line 218: The science behind digit span reveals it's associated more with short-term memory. Why did you consider the Digit Span Test as an attentional task?

This is an issue of wording – short-term memory and attention are not two different categories; instead, attention is part of short-term memory. According to Jonides et al., the digit span test measures the direct-attention performance, which is a crucial part of short-term working memory. We added the reference of this literature in the manuscript (line 224-227).

Reference: Jonides J, Lewis R L, Nee D E, et al. The mind and brain of short-term memory[J]. Annu. Rev. Psychol., 2008, 59: 193-224.

10. Pag.12 line 248: Why you did not include a questionnaire to assess the virtual experience (e.g., cybersickness, motion sickness)? 

We offered an orientation for participants to get comfortable with the VR goggle and ensured immediate stop of the experiment if they feel any discomfort at any time. However, we agreed that including a questionnaire assessing the virtual experience would be a plus to the study.

11. Pag. 12 line 249: Better explain the orientation test and the assumed effect on learning.

Thank you for raising this! Specifically, two number tests (with 3 and 4 digits) were given for orientation, and one example of the AU test was given to demonstrate the criteria of eligible answers. The orientation was assumed to mitigate the learning effect by giving the participant some chances of practicing so that they would not perform poorly at their first few trials due to fresh test. We also added this explanation to the corresponding parts (line 261-267).

12. Pag. 13 line 255: Why 3 minutes?

This study was primarily inspired by Yin et al.’s studies in which the exposure and rest periods were set to be 5 minutes. We discussed the length of VR tasks with Dr. Yin and we shortened the periods to 3 minutes given that the participants of this study will be exposed to five virtual scenes in a single experiment session, which will take approximately 1.5 hours according to our preliminary study. If we set the length of exposure and rest periods to be 5 minutes, the complete experiment will take at least 20 minutes longer. Participants could become tedious, bored, or nervous along the continuous procedure. The unexpected decrease in mental power and attention could potentially become an intervening variable affecting the measurements.

13. Pag. 21 Line 412: Change the expression “biophilic intervention” (e.g., the exposure to different biophilic virtual scenarios)

We first used the expression “biophilic intervention” in the Environmental Simulation section in Materials and Methods, which is a convenient way to refer to the four virtual scenes with biophilic design elements. We added a brief explanation after the first appearance of this expression (line 138-139).

14. Pag. 22 Line 422- 425: rephrase the sentence “we were able to partly confirm our hypothesis about the positive but different impacts of the biophilic interventions from the mental side, but could not confirm the negative impacts of the turbidity based on the current evidence” to make clearer your findings (e.g., in line with our hypothesis, the biophilic intervention was found to have positive effects in promoting subjective wellbeing).

Thank you for raising the confusion. By “partial stress relief,” we meant there was subjective stress relief from the environments, but no objective differences in physiological indicators were observed. We now added the expression to be more precise (line 465-469).

15. Pag. 22 “Stress reaction” section: Why did you not consider emotional competence? There might be noticeable factors impacting the discrepancies between physiological and self-report measures. Given this is interesting data, it might be useful to briefly introduce this argument. 

Thanks for the great suggestion. We now added the brief introduction of emotional competence to discuss this perceived stress relief (line 512-516).

16. Pag. 26 line 521-523: “We used VR technology to bring participants to the Metaverse and explored their responses to the biophilic elements in this digital parallel universe connecting to the real world”. There are two issues in this sentence: first, as highlighted before it is not clear why you talk about Metaverse and not VR alone referring to the proposed experience, second, the “digital parallel universe” might sound a little bit exaggerated. 

We changed to talk about VR alone (as explained in Reviewer 1’s Comment #1). Therefore, the expression “digital parallel universe” referring to the Metaverse is no longer appropriate here and we deleted it (line 566-569).

17. Pag. 27 Line 543: change “VR google”

We revised it accordingly (line 590).

18. Pag. 27 “Limitations” sections: Discuss better limitations, introducing also additional reflections (e.g., add the lack of a measure for cybersickness, possible variables affecting physiological activation such as time, smoking, hunger, the use of self-report measures and evaluations, gender differences).

We do not consider the lack of a measure for cybersickness a weakness of the study (please refer to the response to Reviewer 1’s comment #10). However, we added some other possible intervening variables affecting physiological measures (line 608-610).

Reviewer 2

In this study, the authors aimed to assess the effects of biophilic virtual environments on stress levels and cognitive performance in 30 university students. To do so, the students were immersed in 4 different biophilic virtual environments and 1 control virtual environment using a VR headset. Before, after and during the use of the VR headset, the participants performed different test and answered multiple questions. The study mainly revealed null effects.

Although the topic is relevant and interesting to a broad readership, I believe that the study design, unfortunately, was not adequate to address the authors’ potential research question and hypotheses. I believe that the study is underpowered and the results are not described clearly enough to support the conclusions. In addition, the manuscript fails to address how the findings relate to previous research in this area. The authors should rewrite their Introduction and Discussion to reference the related literature, and the results section to sufficiently describe the conducted analyses. Throughout the manuscript, but especially in the discussion section, the authors describe hypotheses, prior findings and their own findings intuitively rather than comparing it to other literature or providing evidence from prior research. Also, when findings are compared to literature it is mostly to the same one paper (ref 31).

Moreover, the manuscript would also benefit from a proofreading by a native English speaker to improve readability and take out some language mistakes and unclear sentences.

Major issues

1. In general, the introduction is well-structured, but I believe that the purpose of the study, in light of prior studies, is not clear. How does the study relate to other studies, how is it different? What gap in literature do the authors address with their study? And how does that gap lead to the research question and potential hypotheses?

Thank you for raising the confusion! Nevertheless, we actually touched on all the points mentioned here. To make them clearer, we reorganized the Introduction section and stated the research gap (line 66-70), research question (line 89-92) and hypotheses more explicitly (line 102-107).

2. The hypothesis is quite vague. What do the authors mean with different levels of physiological stress reduction and cognitive function improvement? Do the authors expect less stress reduction in some conditions and more in others? Please specify.

In our study, we were implementing various combinations of possible biophilic elements. For example, in “Indoor Green” scene, no windows or outside views was present, but more wall decorations and potted plants were deposited. On contrast, “Outdoor Green” had higher levels of sunlight and outside views. By “different levels” we mean the scenes with different biophilic designs may cause different stress reduction and cognitive improvement. We were not sure which design would excel the non-biophilic environment in stress relief and whether any one of them would achieve that, so we included this variation in environments. As to clarify that this is only an exploratory part in our study, we provided a more detailed explanation in the Introduction to separate it from our hypothesis.

3. Thee research question stated in the abstract does not fully align with the hypothesis mentioned in the introduction section. How do they relate to each other?

Pertaining to Major issue #2, we are sorry that our hypothesis caused the confusion. Our priority goal is to seek evidence for the biophilia hypothesis, thus verifying that the biophilic design can improve human health by lowering the stress level and improving the cognitive function. The “different levels” is an expression to feature our intention of exploring the impact from different designs. The explanation should now be clearer after we changed the wording.

4. Fig 3 is should be adjusted to avoid confusion about the specific study design and procedures. The figure is not easy to read. Please write it out in full in the text and keep only the larger procedural steps in the figure, eg, adding which outcome measures are assessed at which time point and visualization of the different steps a participant takes, would make the procedure more clear.

Thank you for raising these suggestions! The detailed experimental procedures were listed at Experimental Procedure section in the manuscript (line 255-286). We found that the figure is indeed a little hard to follow since we outlined the major steps in the middle, and then the detailed steps for experimental procedure and data analysis sideways. We have now separated the figure into two, one for the experimental procedure (Fig 3) and the other for the data analysis (Fig 4). We abandoned the major steps since it should be intuitive and not much details need to be specified. 

For the figure of experimental procedure, we marked the outcome measurements time points with bold characters to underline them. We also added some visualization icons to better illustrate the procedures taken along the experiment. We put the new figure below for your reference:

We also revised the data analysis procedures and made a new figure for it. Specifically, we added the steps taken in the data cleaning and preparation part, the ANOVA for preference for biophilic patterns and the post-hoc comparisons. We also added some visualization icons to vivify the figure. Please find the new figure below for your reference:

5. I believe that the study is underpowered. The authors perform a lot of statistical tests and have a small sample size. The authors should perform a power analysis and, if needed, also mention the power as a limitation. Also, ss the hypotheses are not clearly stated and the study was not preregistered, it is not clear which statistical tests were planned and which one were exploratory.

Thank you for raising the concern here! Indeed, we suspected that the sample size is too small for this particular study. The power analysis was not performed beforehand due to the insufficient empirical study evidence for similar designs. However, we should perform an afterhand power analysis to more rigorously evaluate the effect. After calculating the effect sizes (resolved in major issue #15, defined as the standardized regression coefficient), we used the function pwr.f2.test in R to calculate the power for the greatest effect size we have, which is the ΔDiastolic Blood Pressure in Trubid Outdoor Green Intervention (0.163). Evaluated at significance level of 0.05, u=4, and v=25, the power is approximately 0.326. This is a low power. If we evaluate the sample size instead, we found that we need at least 79 participants’ data to achieve a power of 0.8. Other parameters witnessed even lower values of effect sizes, thus they would require larger sample size as well. Our final conclusion is that this study is indeed underpowered. We added the power analysis-related part to Materials and Methods (line 323-325), Results (line 386-392, 428-429), and Discussion (line 472-473, 481-485, 507-509, 560-562).

Among the statistical tests, the confidence interval for all physiological and cognitive measurements, and the connection to nature, and ANOVA for biophilic pattern preferences are planned statistical tests which are necessary for examining the biophilia hypothesis. In addition, the linear regression for the learning effect of cognitive testsand ANOVA for baseline measurement of physiological stress reaction are more exploratory. They are to ensure that the participants did not progressively learn to perform better in the cognitive tests, and also the randomization of virtual scenes was successful.

6. The description of the study design does not clarify how this is a crossover study as there are no clear cohorts. Also, between different VR environments there was hardly a wash-out period. How can the authors be sure that the effects in the second, third, fourth or fifth environment are not lingering effects from the first environment? How much physiological effects/stress relief can you still expect after multiple environments? I believe that this is a design choice that severly impacts their results, and this should be mentioned as a limitation.

There are two questions here. One is regarding the defined category of our study, and the other is questioning the absence of a “wash-out period”.

We defined this study as a crossover study because the participants served as their own control in this case. All the participants experienced all five virtual scenes, and the data in the non-biophilic control scene is their own control. In other words, the change of physiological and cognitive indicators between the biophilic scenes and the blank control is calculated for every individual. The linear regression model showed the statistical analysis of those processed changes.

With regard to the second question, we are thankful for your claims as we found out we did not transparently mention the function of the 3-min baseline measurement (3-min quiet sit before each virtual scene). It is the “wash-out period” that is required by the reviewer. We planned out this section to diminish the effect from previous environments and also make baseline measurements for the next scene. We now added more explicit explanations to the paragraph to make it clearer (line 270-279).

7. The authors have not mentioned all the necessary statistics, eg., R2 and p-values in the results section. Also, when reporting the results of a one-way ANOVA, authors need to describe the independent and dependent variables, the overall F-value and the corresponding p-value, and also the results of post-hoc comparisons. This information is missing in the manuscript, but the authors do conclude and describe that there are differences between the VR environments.

Thank you for mentioning these confusions! There is no inclusion of R2 and p-values for the linear regression models of physiological indicators and cognitive functions because we used the 95% confidence intervals instead, which we listed in the tables and texts. We did not include the description about independent and dependent variables because intuitively readers would know that the independent variables are the different virtual scenes, while the dependent variables are the corresponding physiological stress indicators measured for the baseline measurement. For the preferences, the independent variable is the different categories of biophilic patterns, while the dependent variable is the ranking participants gave to the patterns. The overall F-value and the corresponding p-value are listed in supplementary tables S3 and S5. The post-hoc comparison is unnecessary for the baseline measurement of the physiological indicators because ANOVA gave insignificant result. 

However, we did miss the comparison for the preferences. We added a post-hoc Turkey multiple comparisons test to S5 Table. The results showed that there is significant difference between preferences for Visual Connection to Nature and Dynamic & Diffuse Light, and between Visual Connection to Nature and Materials Connection to Nature, but no significant difference between Dynamic & Diffuse Light and Materials Connection to Nature. The description in the corresponding results is modified accordingly (line 451-455).

8. Line 185: I believe that crucial information is missing in the description of the physiological sensors used in order to be able to replicate the study. First, information of, for example, sampling rate for HRV is missing, were artifacts removed, if so, how? How was the data preprocessed, how was the data calculated to use in analyses?

The sampling rate for all the continuous physiological measurements is 128 Hz, which we have added to the description in Physiological Indicators of Stress Reaction (line 197-198). Hence, the artifacts were not removed during the entire experimental process. The conversion of PPG to HR is a built-in function of the ConsensysPro Software so the details are unknown to us. The pre-process of HR to HRV’s formula is included at line 189. The data calculation process is listed in Statistical Analysis in detail. We are not sure what the reviewer is asking for in this inquiry.

9. Line 267: what type of ANOVA, which factors (with which levels) were included in the analyses? Also, did the authors check the assumptions? Which factors were included in the linear regression analyses?

We added the type of ANOVA to the description (line 289-290). No other factors other than the different environments are added to the model because the ANOVA is used to check whether there were differences in baseline measurements. The factors included in the linear regression analyses were included in Statistical Analysis. Please refer to line 299-346 for details.

10. Line 270-272: “In all the analyses of physiological indicators, the measurement before each virtual scene observation served as a baseline parameter that is deducted from the measurement after or during the environmental exposure.” This information is too vague. What do the authors mean with the measurement before? Do the authors work with an interval, did they take an average of that interval? What is considered before, how much time? What do they mean with ‘the measurement ‘after or during the exposure’? How was the physiological data processed before analyses?

Please kindly refer to Major issue #2 and #3 for the answers for this question.

11. Line 326: It is not clear what the authors mean with five baseline measures. Were there 5 cohorts of specific orders of the VR environments? Also, are these the true baseline measures, namely the outcomes measures assessed for the very first time, before any VR environment was experienced, or do these baseline measure refer to the average of each outcome measure before each new VR environment?

Please kindly refer to Major issue #2 and #3 for the answers for this question. Again, we apologize for the confusion of the function of the 3-min quiet sit.

12. Line 393: The authors report the means and SD of the preference scores, but do not report statistical results to support the statement that participants experience higher levels of connections in certain environments. If there are no significant differences between environments, the authors should state this. Also, as the description of the ANOVA analysis uses different wording than in the table, it is not clear which environments are preferred over others.

Thanks for raising the confusion! Besides the means and SDs, we have in fact included the 95% confidence interval of the self-reported connection with nature in Table 6. Additionally, the reviewer might mistakenly mix the ANOVA analysis for the preferences for the biophilic patterns with the results for connection with nature – those are two different things we evaluated.

13. Line 413: The conclusion drawn by the authors is not supported by the results. In the result section, the authors have stated that there were no significant difference and now they suggest ‘partial stress relief’. This does not seem correct to me, nor is it clear what the authors mean precisely. Also, what is the suggested discrepancy between objective stress measurements and subjective feelings?

Thank you for raising the confusion. By “partial stress relief,” we meant there was subjective stress relief from the environments, but no objective differences in physiological indicators were observed. We now think the expression may be clearer (line 465-469). This is also the suggested discrepancy between objective stress reactions and subjective feelings.

14. It is confusing for readers that the authors first describe results, and then state ‘but these results were not statistically significant’. When the results are not statistically significant, the authors should not describe them in that manner.

Thanks for raising this concern! We also recognized this mistake as we reviewed the paragraphs. Stating the differences between interventions and comparing them as if they were significant is not a scientific way to report the results for this study. As a result, we have now separated the results that were significant from those that were not by clearly emphasizing the insignificance at first, and then introduced some trends that were interesting but not significant. We have adjusted the language so that readers would not be misled by the expressions and took insignificant results as significant.

Specifically, we deleted the description of physiological indicators’ change in results, and only discussed an interesting trend for diastolic pressure increase (results line 377-382, and discussion line 518-519). The “positive effect” indicating incorrectly increase in attention was modified to “increase in means” to be more objective and less misleading (line 413-419). Discussions regarding the discrepancy in creativity and physiological indicators were also deleted since they are not significant. 

15. Line 420: The authors state that ‘results were not statistically significant, suggesting small effect sizes’. This is not a correct statement. The authors can calculate effect sizes and report them to make an evidence-based statement.

Thank you for your suggestion! We have now added the calculation of effect sizes in the corresponding parts (line 386-392) and modified the expression in the discussion (line 471-474, 484-485). The effect size was chosen to be expressed as the standardized regression coefficient. All the effect sizes for physiological indicators and cognitive assessments are less than 0.2, which is defined as a small effect size from Cohen et. al. (1988).

16. Figure; the results are not clearly shown in the figure. It is quite a busy figure, showing a lot of information, but it is not clear what ‘position’ refers to, nor what ‘number of appearance or average number test refers to.

Thank you for raising this question! The figures indeed appeared to be confusing since there was not enough description attached to it. We have now updated the figure legends with more detailed explanation. Following is the description for the updated figure legend for this figure, and we hope this will make the information clearer.

Fig 5. Ordering effects in cognitive tests. Position A-E stands for the appearance order of the environments. A means the environment that appears first in one individual experiment procedure, B means the second environment, etc. “Number of Appearance” measures how many times the environment is ordered at this position among all the experiments. The green line is “Average Number Test Score” which is the mean score for all the number tests done after each scene at the position. For example, the dot at position A stands for the average score for all the number tests done after the first environment.

17. Line 480; The authors make incorrect claims about the attention improvement, as they stated prior that there was a learning effect. (line 390)

Please kindly refer to Reviewer 2’s major issue # 14 for this problem.

18. Line 490: With this statement the authors show that their study, unfortunately, was not ideally designed for their research questions. I believe that the learning effect is independent of the order of the VR environments. The learning effect occurs due to repeated testing, up to five times. You can expect that even after 1 cognitive test participants will perform better the next time, and here the authors have repeated it 5 times in 90 minutes due the pre and post assessment sessions (if I have understood the study correctly). Authors should include information on test-retest reliability and repeated assessment. It was also not clear to me, until line 565, that there was no baseline cognitive measure. The authors should clarify this in the study design figure.

Thank you for raising the confusion! Indeed, the learning effect should be independent from the order of environments. However, we tried to use the randomized order of environments to counteract the learning effect. This is because if there are significant improvements for the number test in one particular intervention, the results would not be caused by the learning effect since its appearance position in each experiment will be different. Therefore, we are suggesting here that we are surprised that the randomization did not work. We believe the insignificant results were due to a combined effect from both the learning effect and the unbalanced order. Unfortunately, we did not have pilot measurements that could indicate the test-retest reliability for the number test, and that should be included in the future study.

Regarding the baseline measurement for cognitive assessment, we did not include it due to potential learning effect because this would require doubling of the number of practices for each individual. We are sorry for the missing of explanation for it, and we added it to the end of Cognitive Function Assessment section (line 248-252). It is not included in the study design figure because the illustration would be too wordy and would possibly cause more confusion, as was raised by the reviewer at issue #4.

19. In the ‘strengths and limitations section’ the authors make some statements without backing them with evidence. For example, as a 3rd strength the authors state that the comparison between physiology and self-reported stress levels is a strengths, but the authors have not formally compared this (ie. with correlation analysis), as they only described both results. Also, the authors state that they provided a very immersive experience, but as they did not formally assess sense of presence/immersion (ie with IPQ), this is again an unfunded statement.

Though we did not formally assess the sense of immersion, studies showed the 3D models in the head-mounted display could provide a more immersive experience than 2D pictures. We now added some relevant literature to the manuscript (line 586-588).

20. The statistics are not reported correctly, which makes the results section a bit confusing to read. The authors on multiple occasions start describing the results to then state that the results were not significant. This is confusing for the reader. The conclusions are not supported by the results, the authors describe results as if they were significant and the discussion section would benefit from a clear focus. Also, the authors report multiple findings in the discussion section that were not described in the results section (eg line 494-499: order of the presented environments).

Please kindly refer to Reviewer 2’s major issue #14 for the first problem here and Reviewer 2’s minor issue #30 for the second problem.

21. Line 413: The authors mention new information that was not mentioned in the methods section nor results section, namely the removal of anomalies in the data. The authors should include this analysis step in the methods and analysis section and clarify what is considered as an anomaly.

Thank you for raising the question! The anomaly was referring to the extremely large difference compared to other data points in the pre-post change physiological measurement, and for the data in cognitive assessment. They were sieved out by boxplot function. We added the specific explanation to the Materials and Methods section (line 307-310).

22. Line 444: Seen as there are no significant findings concerning the physiological outcome measures, the authors should not interpret the results as if there were significant differences between these groups.

Please kindly refer to Reviewer 2’s major issue #14 for this problem.

Minor issues

23. Considering that the primary outcome measures of the study are the physiological measures (though not stated explicitly), the authors should consider including more information in the introduction section on how physiological measures are linked to stress, as this information is currently missing.

We explained how each physiological indicators are linked to stress in the Physiological Indicators of Stress Reaction section (line 170-171, 183-186). We did not include the detailed information in the Introduction section because it would make the Introduction section not concise and hard to navigate.

24. Line 135 and 137. It is not clear what references 35 and 31 actually refer to, or what evidence they provide. Please clarify this in the text. Describe the 3 biophilic design patterns and explain why they are suitable for short exposure time, and what is considered a short exposure time. Perhaps it might even be better to already explain this in the introduction section.

Primarily, we have described the three biophilic patterns in the first paragraph of the Environmental Simulation section as follows:

“Specifically, the patterns of "Visual connection to nature" and "Dynamic and diffuse light" were combined to represent Nature in the Space, which included potted plants, trees, sky, clouds, and access to natural light and shadow. We used the pattern of "Material connection with nature" to represent Natural Analogues, including wooden floors and ceilings.”

Secondly, the reviewer referred to the two references in the following sentence.

“In this study, we chose three biophilic design patterns [ref1] for the virtual classroom design because (1) they relate to indoor classroom design, (2) they can be vividly simulated in VR, and (3) they are suitable for short exposure time [ref2].”

The first reference refers to the paper where the three biophilic design patterns that we chose are introduced and analysed. We now added the explanation before the reference. The second reference refers to the reasons for choice given by the study which chose the same three biophilic design patterns.

Last but not least, a few minutes (e.g., 5 minutes) is considered a “short exposure time.” The reason why the three biophilic design patterns (i.e., visual connection to nature, dynamic and diffuse light, and material connection with nature) are suitable for short exposure time is because previous studies show that a few minute of exposure to the biophilic elements (e.g., green space) in those patterns (e.g., visual connection to nature) could have significant impacts on people’s stress level and cognitive functions. We now added this explanation.

25. Line 201: Stress level rating, what did the 1 refer to, what did the 5 refer to? 

We added the explanation: 1 refers to “not stressful” and 5 denotes “extremely stressful (line 205-206).

26. Line 214: Cognitive tests delivered verbally might be a limitation as they were not validated as such. It is not clear to me how the tests were delivered. If the participants were still seeing the VR environment, doesn’t this create some sort of sensory overload?

Thanks for raising the confusion! We have described how the tests were delivered in the Cognitive Function Assessment section (line 217-219). We conducted both tests verbally to avoid the potential visual interruption caused by the switch of interface in VR goggle. This is to maintain the participants in the VR scene while taking the tests. This may create some sensory overload, but they were asked to sit still to reduce this potential effect. Moreover, we agreed that conducting the AU test verbally might be a limitation (note that verbal backward digit span task is by convention a verbal task) and we added that in the Strengths and Limitations section (line 610-614).

27. Line 227: The authors state that ‘divergent tasks are more complex cognitive assessments and relate to more creativity’, without providing a reference for this claim. Please do so.

Thank you for pointing that out and we now provided a reference (line 234).

28. If participants are allowed to move during the physiological assessments (during VR use), how did you correct for movement artefacts?

Unfortunately, we were not able to control the movement artefacts during the experiment. Our device indeed collected data regarding the 3D position axis changes during the recordings, but including the correction from this data would likely randomize the data more because different participants followed different paths of exploration.

Instead, we tried to minimize the movement of the recording equipment during the experiments. We asked during the orientation period that the participants should move their arm (the one with the Shimmer 3+ Unit) minimally. The position is maintained when they stood up and sat down. We know this was not the best solution to controlling the artefacts, but we also would like the participants to have more immersive experience. Therefore, we chose a balanced decision by making them move but keeping the arm still.

29. Line 261: Did the online check-out survey consist of validated questionnaires? Did the authors use open-ended or close-ended questions?

We attached the online check-out survey in our new submission (S1 Survey). We used close-ended questions: e.g., Yes/No, rate from 1 to 10, rank the top 3 preferences, etc.

30. Line 269: how was the order of the randomized virtual environments determined? Were there groups of the same order? The authors should provide information on how many times each environment was experienced in which place in the order? If this was completely random and, for example, environment 1 was only 2 times in the second place, whereas environment 3 was 20 times in the second place, then how can you assess potential order effects? As this (new) information is later addressed in the discussion section, it should be addressed first in the methods and results section.

Pertaining to this problem, we first would like to confirm that the virtual scenes were indeed randomized. It was done by using the function “random.shuffle()” in the “random” package in python. We defined a list of five numbers and randomly shuffled the numbers to get the order of virtual scenes at certain positions. 1, 2, 3, 4, and 5 represent non-biophilic, biophilic-indoor, biophilic-outdoor, biophilic-turbidity, and biophilic-combination correspondingly. The code was copied below for reference. 

order = [1,2,3,4,5] 

random.shuffle(order) 

print(order) 

The time that each virtual scene was experienced in which place in the order is provided in Fig 5. Indeed, we observed pseudo-randomization effect for our experiments. That is what we are trying to discuss at line 433-438. It was also mentioned in Reviewer 2’s comment #20. We have now added the description to corresponding parts in Results and changed the expression in Discussion (line 539-544).

31. The statistical analysis section does not describe which statistical program is used, what the level of significance is, which post-hoc corrections were applied. Also, no information on checking of assumptions is provided.

We added the description of statistical programs used, and all the information to the beginning of Statistical Analyses section (line 289-292). However, we are not sure what the “assumptions” the reviewer mentioned here is.

32. The methods section would benefit from a separate ‘outcome measures’ section detailing the different outcome measures and the time points at which they are assessed. The authors could do this visually in the procedure figure.

Please kindly refer to Reviewer 2’s major issue #4 for the response.

33. The results section mentions both diastolic and systolic blood pressure, whereas the difference between both or relevance of assessing both is not explained in the methods section. Please do so.

We now added the definition of two numbers and explained why we included both in our analysis (line 173-177).

34. Stress reactions section: For clarity, the authors should state upfront that the results were not statistically significant instead of in the end. The authors should do this throughout the results section and discussion section.

Please kindly refer to Reviewer 2’s major issue #14 for this problem.

35. Table 3: How was the self-reported stress level per group calculated? As these are negative numbers, are these difference scores? If so, please mention this in the results section, as well as in the methods section.

It is the difference in the self-reported stress levels between the biophilic interventions and the control non-biophilic scene – we mentioned this in Table 3’s title. We rephrased the sentences to make them clearer and added the explanation in methods section (line 206-208).

36. Line 362: Was there a significant difference in average (pre-to-post changes in) stress levels? Or is this a descriptive analysis?

According to the 95% confidence interval provided in Table 3, there was a significant difference in the self-reported stress levels between the biophilic interventions and the control non-biophilic scene – this is exactly what we wrote in the original sentence but we rephrased it to make it clearer.

37. Line 435: I do not agree with the authors that the ‘Outdoor Green’ is an open space. It is still a classroom, so rather considered as a closed space. The authors should add literature on what is considered open and closed space and how this feature relates to their findings, if the authors want to mention this feature. The mentioned reference does not support the statement, or is not good comparison material, seen as in that study there was a noticeable difference in the scale of the space shown. In this study, the scale of the virtual rooms is the same, as it is the same room.

We agreed with the noticeable difference in the scale of the space in this study and the one we referred to, meaning we cannot classify our scenes as “open space” and “closed space” as Yin et.al’s study. Therefore, we took off this part from the manuscript.

38. Line 440: Can the authors clarify what they mean with ‘subjective feelings of nature’ and change it accordingly in the manuscript?

The subjective feeling of nature means the self-report scores of connection with nature (score 1 ~ 10) and we introduced this in the Experimental Procedure section (line 282-284).

39. Lines 518-519: The authors should elaborate on the why and how of this statement. What do they refer to with their reference?

The reference points to “the inclination that reproduces the physical world in the virtual world.” (line 566)

Additional minor remarks

40. Line 36, 77: Grammatically incorrect sentence. I believe that the word ‘that’ is missing.

We revised it accordingly.

41. Line 54: decreased instead of decreasing

We revised it accordingly.

42. Line 76: Please rephrase. This sentence is quite a bold statement that assumes causation based on completely unrelated studies.

Thanks for the suggestion. However, we did not assume any causation here; instead, we were saying the increases in smart devices usage and screen times are found to be associated with (1) a variety of stress-related symptoms and (2) students’ worse academic performance.

43. Line 116: ref 34 is not a correct reference for the statement that is made. The authors refer to another article of one of the authors in which they state that using a cross-over design would provide a larger power with the same number of participants, but that study did not actually assess this. Please use a correct reference to indicate your choice for this design. Also, was a power calculation conducted beforehand? If not, please provide a power calculation in the manuscript and , if the power is insufficient, add this information in the limitations section.

Indeed, the original reference only stated that the crossover design can achieve the same power with few participants, while they did not reference corresponding literatures. We have now added a supporting empirical study that used meta-analysis to show that the estimated increase in power and fewer sample size requirement are reasonable (line 98-101). 

Please refer to Reviewer 2’s major issue #5 for the second problem.

44. Line 126 should be inclusion criteria, not including criteria.

We revised it accordingly.

45. Line 157: should state NVIDIA instead of NVIDA

We revised it accordingly.

46. Line 169: should state refresh rate instead of fresh rate.

We revised it accordingly.

47. Line 176: What do you mean with ‘obtained by calculation’, can you please specify?

In general, we started each section with a brief summary, followed by the detailed descriptions. That is why we only wrote “obtained either by biomonitoring sensors or calculation” in the first paragraph under the “Physiological Indicators of Stress Reaction” section. We speficied our calculations in the third paragraph of the section (line 179-198).

48. Line 210: I do not fully understand this sentence. Do the authors state that VR has the higher potential to boost cognitive function than traditional methods or does VR provide a better method to assess cognitive function that traditional methods? The wording is not clear on this. Also, the study that is referred to is quite old in terms of VR research. I believe there is more recent work that can provide some background on using VR for cognitive assessment. Also, if referring to VR for cognitive assessment, why do the authors then choose verbal tests? Are the tests that were used validated to use them in this manner? If there is no VR assessment of cognitive function in the study, there is no need to provide background information on the matter.

Firstly, by “Empirical research suggested that VR has the higher potential to boost cognitive function assessments than traditional methods,” we meant VR can potentially facilitate better cognitive function assessments than traditional methods. We modified our wording accordingly (line 211-212).

Secondly, please refer to our response to Reviewer 2’s comment #26 regarding the delivery method of the cognitive tests. Besides that, regarding using VR for cognitive assessment in this study, we used two validated cognitive tests to assess the potential impacts of the biophilic elements in the virtual scenes displayed in VR on participants’ attention and creativity. In other words, VR is used to affect participants’ cognitive functions, not deliver the cognitive tests.

Thirdly, as suggested, we added references of recent work on virtual reality tests of cognition. Although some referred studies conducted the cognitive tests using VR (vs. conducted the tests verbally in this study), the point is that VR can potentially facilitate better cognitive function assessments.

49. Instead of Characteristics from the 30 visit, which is confusing wording, the authors might want to use ‘circumstantial factors’ or something similar.

We changed it to “characteristics of the participants”.

50. Why are the participants divided into two age groups; 18-19 and 20-21? Please provide mean age and SD in the table, as was described in the text.

We deleted the age groups since it is not necessary to divide them into two age groups. But we have provided the mean age and SD in the table, please refer to line 366 for the information.

---

## [Decision Letter · Decision Letter 1]

6 Jul 2023

PONE-D-22-35023R1Biophilic classroom environments on stress and cognitive performance: A randomized crossover study in virtual reality (VR)PLOS ONE

Dear Dr. Ji,

Thank you for submitting your manuscript to PLOS ONE. After careful consideration, we feel that it has merit but does not fully meet PLOS ONE’s publication criteria as it currently stands. Therefore, we invite you to submit a revised version of the manuscript that addresses the points raised during the review process.

Reviewer 3 suggested minor revision. I encourage Authors to include them to proceed with the process. Especially I agree that some aspects were addressed in responses to previous revisions but not implemented as changes in the manuscript. 

We look forward to receiving your revised manuscript.

Kind regards,

Stefano Triberti, Ph.D.

Academic Editor

PLOS ONE

Journal Requirements:

Reviewers' comments:

Reviewer's Responses to Questions

**Comments to the Author**

1. If the authors have adequately addressed your comments raised in a previous round of review and you feel that this manuscript is now acceptable for publication, you may indicate that here to bypass the “Comments to the Author” section, enter your conflict of interest statement in the “Confidential to Editor” section, and submit your "Accept" recommendation.

Reviewer #3: (No Response)

2. Is the manuscript technically sound, and do the data support the conclusions?

Reviewer #3: Yes

3. Has the statistical analysis been performed appropriately and rigorously? 

Reviewer #3: Yes

4. Have the authors made all data underlying the findings in their manuscript fully available?

Reviewer #3: Yes

5. Is the manuscript presented in an intelligible fashion and written in standard English?

Reviewer #3: Yes

6. Review Comments to the Author

Reviewer #3: Reviewer 1 and 2’s concerns were mostly adequately addressed; Some minor aspects still need authors’ attention.

- Line 90: the sentence “we used VR and wearable biomonitoring sensors to quantify the impacts of both positive and negative factors in built virtual environments on short-term health of university students” is not clear. What do the authors refer to by the term "short-term health"? Both stress and cognitive function? It would be better to clarify.

- It is suggested to add, in the introduction section, present studies in order to clarify what kind of link has been found in the literature between stress, creativity and exposure to natural environments (both real and virtual). A few lines are needed.

- Reviewer 1 noticed that Verbal Backward Digit Span Task (Number test) is a test for working memory and not for attention; Authors responded citing literature that conflates the two constructs. This is satisfying; however, it seems that nothing has been changed in the manuscript in this regard. Authors should add this specification and the related citation to the manuscript, not only in Response to Reviewers.

- We recommend adding in the discussion or limit section studies that support the role of interaction with the virtual environment in stress reduction (see, for example, Liszio, S., & Masuch, M. (2019). Interactive immersive virtual environments cause relaxation and enhance resistance to acute stress. Annu Rev Cyberther Telemed, 17, 65-719). In fact, it may be possible that participants' lack of interaction with your study material reduced the possibility of achieving the expected results regarding stress reduction through other measurements as well.

- Line 465: "After removing the anomalies, no significant differences were found for physical measurements of physiological indicators." Also emphasize here that no differences were found at the cognitive level.

7. PLOS authors have the option to publish the peer review history of their article (what does this mean?). If published, this will include your full peer review and any attached files.

Reviewer #3: No

---

## [Author Response · Author response to Decision Letter 1]

17 Aug 2023

Thank you for pointing this out. We replaced two missing references (old ref. 8 and 66) with more recent and most-cited ones (new ref. 10 and 68).

1. Line 90: the sentence “we used VR and wearable biomonitoring sensors to quantify the impacts of both positive and negative factors in built virtual environments on short-term health of university students” is not clear. What do the authors refer to by the term "short-term health"? Both stress and cognitive function? It would be better to clarify.

The reviewer’s question for clarification is a clinically relevant point, as we should use more specific terminology. Short-term may mean different time scales according to different disciplines. In our language, we focused on non-chronic diseases, meaning that the etiology does not take many years to develop.

By “short-term health,” we mean immediate physiological and cognitive responses (i.e., physiological & perceived stress levels and cognitive tests performance) to the virtual scenes during the 90-minute experiment session. Furthermore, we also clarified “short-term” with added explanation that it refers to immediate impacts on health within the day, or min/hours following exposure (line 101-102, 115-116).

2. It is suggested to add, in the introduction section, present studies in order to clarify what kind of link has been found in the literature between stress, creativity and exposure to natural environments (both real and virtual).

The link between stress/cognitive function (including creativity) and exposure to real natural environments is discussed in the Introduction section as follows:

“Population epidemiology studies and experiments documented exposure to outdoor nature (e.g., greenspace) can positively affect human health and well-being in multiple ways: reduced stress level, improved mental health and cognition, lower mortality rates [12-16], and enhanced immune functions [17-20].” (line 70-74) and “There is emerging evidence that biophilic design in simulated environment can yield health benefits. For example, indoor plants could be conducive to stress-reduction and attention restoration [22, 23]” (line 81-83).

The link between stress/cognitive function (including creativity) and exposure to virtual natural environments is discussed in the Introduction section as follows:

“Combined biophilic design elements using virtual reality (VR) by Yin et al. [30] demonstrated that bringing nature into virtual indoor workspace has clear benefits to the health outcomes, including physiological stress reductions and cognitive function (attention and creativity) improvements.” (line 88-92).

In terms of the link between stress and creativity, we did not address that in the manuscript since it is not part of our research question.

3. Reviewer 1 noticed that Verbal Backward Digit Span Task (Number test) is a test for working memory and not for attention; Authors responded citing literature that conflates the two constructs. This is satisfying; however, it seems that nothing has been changed in the manuscript in this regard. Authors should add this specification and the related citation to the manuscript, not only in Response to Reviewers.

We revised the manuscript accordingly and added the citation (line 238-239). We may have had an oversight changing many aspects in the previous version. We thank the reviewer for their fastidiousness in ensuring our manuscript is scientifically sound with respect to test and corresponding cognitive domains. 

4. We recommend adding in the discussion or limit section studies that support the role of interaction with the virtual environment in stress reduction (see, for example, Liszio, S., & Masuch, M. (2019). Interactive immersive virtual environments cause relaxation and enhance resistance to acute stress. Annu Rev Cyberther Telemed, 17, 65-719). In fact, it may be possible that participants' lack of interaction with your study material reduced the possibility of achieving the expected results regarding stress reduction through other measurements as well.

Absolutely, and it is essential that we address the limitations appropriately. As you know, our study participants were mainly students, and this population is healthy but also has many virtual inputs throughout the day. Certainly, if there are more interactions in the virtual space, it may change the dynamic of the biological response measurements. To capture this point, we added this argument in the Strengths and Limitations section as a recommendation for future VR studies (line 642-645).

5. Line 465: "After removing the anomalies, no significant differences were found for physical measurements of physiological indicators." Also emphasize here that no differences were found at the cognitive level.

We emphasized this point and revised it accordingly (line 483). Although, we think our finding can instigate more studies as better digital biomarkers technologies become available. At this point, subjective finding is a start, and the lack of physiological response finding may be due to ability to detect differences, or because of sample size. We thank the reviewer in making sure that this point comes across properly.

---

## [Editor Report · Decision Letter 2]

29 Aug 2023

Biophilic classroom environments on stress and cognitive performance: A randomized crossover study in virtual reality (VR)

PONE-D-22-35023R2

Dear Dr. Ji,

We’re pleased to inform you that your manuscript has been judged scientifically suitable for publication and will be formally accepted for publication once it meets all outstanding technical requirements.

Kind regards,

Stefano Triberti, Ph.D.

Academic Editor

PLOS ONE
---

## [Editor Report · Acceptance letter]

13 Oct 2023

PONE-D-22-35023R2 

Biophilic classroom environments on stress and cognitive performance: A randomized crossover study in virtual reality (VR) 

Dear Dr. Ji:

I'm pleased to inform you that your manuscript has been deemed suitable for publication in PLOS ONE. Congratulations! Your manuscript is now with our production department. 

Kind regards, 

on behalf of

Prof. Stefano Triberti 

Academic Editor

PLOS ONE